# xVal: A Continuous Numerical Tokenization for Scientific Language Models

## Abstract

Due in part to their discontinuous and discrete default encodings for numbers, Large Language Models (LLMs) have not yet been commonly used to process numerically-dense scientific datasets. Rendering datasets as text, however, could help aggregate diverse and multi-modal scientific data into a single training corpus, thereby potentially facilitating the development of foundation models for science. In this work, we introduce xVAL, a strategy for continuously tokenizing numbers within language models that results in a more appropriate inductive bias for scientific applications. By training specially-modified language models from scratch on a variety of scientific datasets formatted as text, we find that xVAL generally outperforms other common numerical tokenization strategies on metrics including out-of-distribution generalization and computational efficiency.

## 1 Introduction

### 1.1 Though LLMs struggle to encode numbers, they are still being adapted for data analysis

LLMs have historically struggled to solve simple arithmetic problems such as multi-digit multiplication (Dziri et al., 2023) and have a tendency to "confabulate" answers (OpenAI, 2023; Frieder et al., 2023). Standard LLM tokenization schemes struggle to capture the precise quantitative properties that distinguish numerical data from other natural language inputs, e.g. the magnitude and continuity of numbers (Testolin, 2023; Choi, 2021). Recent work exploring Chain-of-Thought reasoning in LLMs has shown improved performance on common sense reasoning tasks such as arithmetic or mathematical word problems (Nye et al., 2021; Wei et al., 2023; Liu & Low, 2023; Imani et al., 2023), but such methods have limited applicability in the analysis of scientific datasets comprising mostly numbers.

Despite these fundamental challenges in representing numbers in text format, some recent work has suggested that LLMs can be applied to certain numerical datasets with surprisingly strong results. Gruver et al. (2023) and Jin et al. (2024) found that LLMs such as GPT-3 (Brown et al., 2020) and LLaMA-2 (Touvron et al., 2023) could match or exceed purpose-built models in a zero-shot context, i.e. with no fine-tuning, on a number of time series prediction tasks. This performance is contingent on pre-processing the input data to follow a particular format. First, numbers are scaled such that the $\alpha$-percentile (where $\alpha$ is a hyperparameter) of the dataset is equal to one. Next, a certain level of precision is fixed, and decimal places are omitted or replaced with the digit 0 as appropriate. Finally, numbers are represented with spaces between each digit to force the GPT models to tokenize each digit individually, which is known to be more effective than byte-pair encoding for numerical computations (Liu & Low, 2023). An example of this pre-processing for 2 digits of precision is: $0.123, 1.23, 12.3, 123.0 \rightarrow 1\ 2\ ,\ 1\ 2\ 3\ ,\ 1\ 2\ 3\ 0\ ,\ 1\ 2\ 3\ 0\ 0$.

While the textual inputs to LLMs can be pre-processed somewhat to improve out-of-the-box numerical predictions, additional progress will likely be limited without making modifications to the fundamental architecture of the language models themselves to better represent numerical data.

## 1.2 Language model architectures can be modified to better handle numerical data

Recent work has explored several potential improvements for encoding numerical information as inputs to language models (see Thawani et al. (2021) for a review). For instance, numbers can be encoded digit-by-digit, in scientific notation format, or in base-10 format. (Jiang et al., 2020) map numbers onto a finite set of "prototype numerals", while (Sundararaman et al., 2020) enforce constraints such that the cosine distances between the embeddings of numbers reflects their actual mathematical distance. Transformers that use such encodings have been shown to successfully solve some mathematical tasks including matrix multiplication (Charton, 2022).

Despite these improvements, many challenges remain unresolved. Language models are known to exploit shortcuts and spurious correlations in the data (Tu et al., 2020; Liu et al., 2022; Dziri et al., 2023) and still struggle with interpolation and out-of-distribution generalization in mathematical problems and in scientific domains (Grosse et al., 2023; Anil et al., 2022).

## 1.3 Scientific data analysis often features continuous functions

Functions appearing in scientific domains are often continuous or smooth, with certain exceptions such as points of criticality. Transformer architectures applied to vision and audio domains (e.g., Dosovitskiy et al., 2020; Garg et al., 2022) typically treat numbers continuously without tokenization (see however Copet et al., 2023; Chen et al., 2020), but these models typically require highly structured inputs and cannot be applied to datasets with arbitrary sequences of text and numbers. In contrast, when encoding numbers as text, LLMs are inherently discontinuous in both the encoding and decoding stages. While discrete models can (and do) learn to approximate continuous functions (d'Ascoli et al., 2022), this can be more challenging and less sample-efficient compared to models that have continuity built-in by construction, as in many non-parametric regression models (Wasserman, 2006).

In this work, we introduce XVAL, an inherently continuous method of encoding numerical values in scientific language models, i.e. custom language models that are dedicated to scientific data analysis. By encoding the magnitude of numerical values multiplicatively and orienting them in a learnable direction within the embedding space, XVAL substantially changes how numbers are processed and interpreted by transformer architectures. This leads to an encoding scheme with a single vocabulary element that also encodes every number as a single token. XVAL is therefore both token-efficient and has minimal vocabulary footprint.

Coupled with a modified number-inference paradigm, XVAL allows a transformer model to be continuous (or smooth given smooth non-linearities) when considered as a map between the numbers of the input string and those of the output. We expect that this leads to a better inductive bias when the functions being approximated are continuous or smooth. We evaluate XVAL on two examples of scientific datasets and compare its performance with existing number encoding schemes. We demonstrate that XVAL is both more computationally-efficient and exhibits better out-of-distribution interpolation properties than these baselines.

**Our contributions**

- We introduce XVAL, a novel approach for encoding numerical values in Large Language models specialized for scientific data analysis. Compared to existing number encoding schemes, XVAL is both token-efficient (every number is encoded as a single token) and has a minimal vocabulary footprint (a single number token).

- We introduce a modified number inference scheme that, when used in conjunction with XVAL, renders transformer models continuous as a function of the numerical values appearing in the text.

- We evaluate XVAL and a number of existing number encoding schemes on two examples of scientific datasets. We demonstrate that XVAL consistently provides better interpolation properties and is more compute-efficient than prior work.

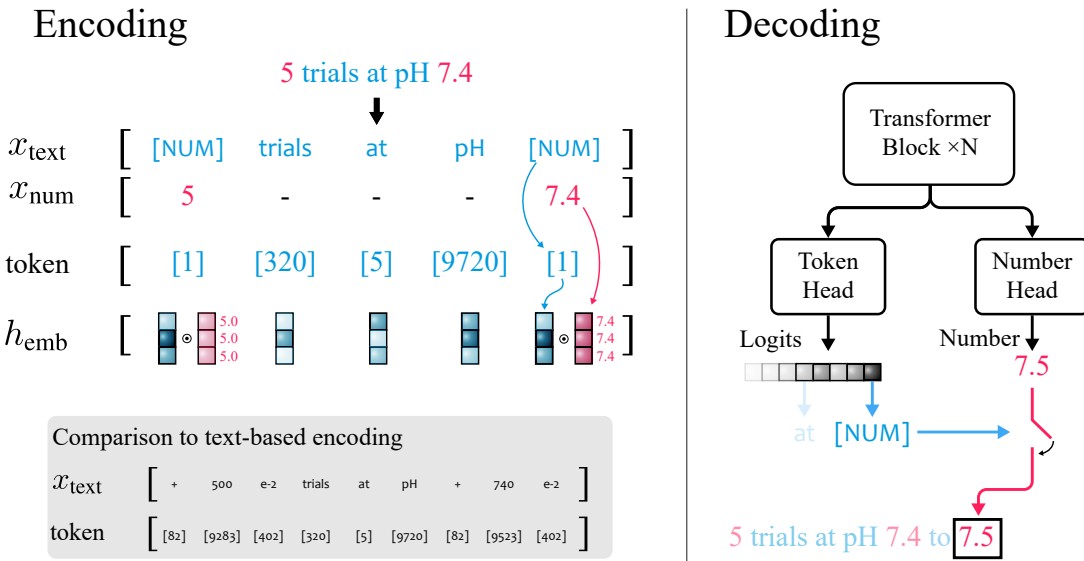

Figure 1: A simplified example illustrating the xVal number encoding and the modified number inference paradigm. On the left, xVal is contrasted with the P1000 text-based numerical encoding scheme. On the right, we illustrate how numbers are addressed within the decoder.

## 2 Methods

### 2.1 xVal: A Continuous Number Encoding

Instead of using different tokens for different digits or composite numbers, xVal embeds numerical values directly along a specific learnable direction of the embedding space. A diagram of this procedure can be seen in Fig. 1. Specifically, given a string input $x$ comprising both numbers and text, we first parse $x$ to extract all the numerical values and collect them in a separate list $x_{\text{num}}$. We then construct a new string $x_{\text{text}}$ by replacing all numbers in $x$ with a designated token [NUM] that acts as a placeholder for numerical values. We tokenize and embed $x_{\text{text}}$, arriving at $h_{\text{text}}$. We then multiply the embedding of each appearance of the [NUM] token with its associated numerical value in $x_{\text{num}}$. This process can be done efficiently by defining a new list $h_{\text{num}}$ by scattering $x_{\text{num}}$ to have the same length as the tokenized $x_{\text{text}}$ and inserting a 1 for any token other than [NUM]. The final embedding of the sample is $h_{\text{emb}} = h_{\text{num}} \times h_{\text{text}}$, which is then fed to the transformer trunk.

This encoding process can be performed both for masked language modeling (MLM) and auto-regressive (AR) generation. During training, in cases where MLM is used, we simultaneously mask both $h_{\text{text}}$ and $h_{\text{num}}$, i.e., if the token being masked is a [NUM] token, we replace the corresponding number in $h_{\text{num}}$ with 1.

Continuous embeddings have been previously proposed for use in attention mechanism in the context of speech recognition (Chorowski et al., 2014).

**Representing a wider range of numerical values** Scientific datasets can cover a vast range of numerical values depending on the sub-field or chosen units. To enhance the ability of xVal to simultaneously represent numbers across a wide range of values, we include the ability to assign $k \in \mathbb{N}$ numerical embeddings centered around different orders of magnitude $\mathcal{O}(10^i)$, where $i \in \mathbb{N}$ and $i \leqslant k$:

$$\sum_{i \in [-k,k]} \tanh(x \cdot 10^i) \cdot [\text{NUM}_i] \tag{1}$$

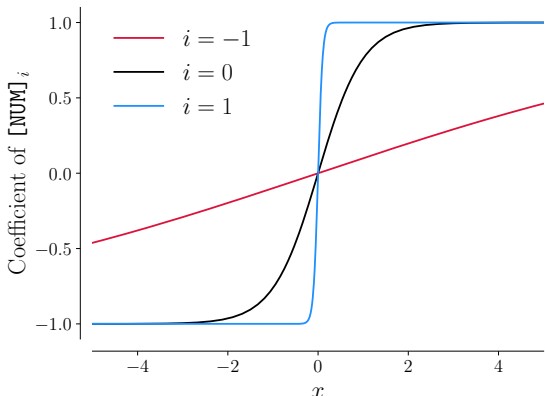

Figure 2: The coefficients of each `[NUM]` embedding vector in Equation 1 are shown for an example of xVAL with $k = 1$, which spans three orders of magnitude. This illustrates how the default xVAL embedding ($i = 0$), which is primarily sensitive to values of $\mathcal{O}(1)$, can be supplemented with additional `[NUM]` embeddings sensitive to $\mathcal{O}(10)$, i.e. $i = -1$, and $\mathcal{O}(10^{-1})$, i.e. $i = 1$. This paradigm can be extended to wider dynamic ranges.

An example illustration of the dynamic range covered by xVAL for $k = 1$, spread across three different numerical embeddings, is shown in Fig. 2. The default behavior of xVAL is to use a single numerical embedding vector, i.e. $k = 0$.

## 2.2 Numerical value inference

xVAL defines an embedding that is continuous in the numerical values of the input. However, if we use a multi-class classification task as our output and training algorithm, the model as a whole will not be end-to-end continuous when considering the map from the input numbers to the output numbers. For this reason, we treat numbers separately at the output layer, as illustrated in Fig. 1.

As is standard practice in transformer-based language models, we define a token head that outputs a probability distribution of the tokens of the vocabulary. However, since our formalism replaces numbers with the `[NUM]` token, this head does not carry any information about the number value. We therefore introduce a new number head with a scalar output. We train via Mean Squared Error (MSE) loss, for $k = 0$, or Normalized Mean Squared Error (NMSE) loss, for $k > 0$, to recover the numerical value associated with each instance of the `[NUM]` token. For any input, we first look at the output of the token head. If the generated token is the `[NUM]` token (or in the span of the `[NUM]`$_i$ tokens, if $k > 0$), we then look at the number head to fill in the value for this token. As shown in Section 3, since the transformer is now end-to-end continuous when inferring numerical values, it performs better when interpolating to previously unseen values.

## 3 Experiments

In this section, we evaluate the performance of xVAL and highlight its strengths and weaknesses compared to existing numerical encoding algorithms. In particular, we look at two datasets: a dataset of global temperature data and a dataset of planetary orbit simulations. In Appendix B, we also investigate arithmetic operations.

For our transformer models, we use an architecture based on GPT-2 (Radford et al., 2019). Details of our specific architecture are included in Appendix C). We explore the effects of various architectural design choices in Appendix D.4.

**Comparison with other number encodings.** We compare the performance of xVAL with four other number encodings, following the notation of Charton (2022). In these encodings, numbers are first processed

Table 1: Comparison of XVAL with four other number encodings. XVAL is more token-efficient and has a minimal vocabulary footprint. Vocabulary size differs from Charton (2022) because we only consider exponents from `1E-8` to `1E+8`.

| Encoding | Tokens | $-6.02 \times 10^1$ | Tokens per number | Vocabulary Size |
|----------|--------|---------------------|-------------------|-----------------|
| P10 | {±, d, E±d} | [-, 6, 0, 2, E-1] | 5 | 28 |
| P1000 | {±, ddd, E±d} | [-, 602, E-1] | 3 | 918 |
| B1999 | {±ddd, E±d} | [-602, E-1] | 2 | 1816 |
| FP15 | {±ddd E±d} | [-602 E-1] | 1 | 28800 |
| XVAL | {[NUM]} | [NUM] | 1 | 1 |

into the format ±`ddd` `E`±`d`. The encodings are then determined by which parts of this format are encoded as single or multiple tokens. These range from encodings with limited vocabulary size but high number of tokens per number, leading to longer encoded sequence lengths (e.g., P10), to those with very large vocabulary footprints but only one token per number, leading to shorter encoded sequence lengths (e.g., FP15). XVAL provides a minimal vocabulary footprint and uses just a single token per number, leading to the shortest sequence lengths. A summary of these encodings and an example can be seen in Table 1.

Number encodings that do not lead to a fixed number of tokens for all numbers (e.g., learned Byte Pair Encoding (Gage, 1994) used in GPT-2 (Radford et al., 2019)) can lead to erratic behaviors where the transformer learns spurious correlations that exist between the length of the encoded numbers in the dataset. An example of this type of behavior is shown in Appendix D.3.

## 3.1 Temperature forecasting

As an example of real-world scientific analysis, we look at the task of temperature forecasting. In this experiment, we construct a dataset as a subset of the ERA5 global climate dataset (Hersbach et al., 2020). For simplicity, we only focus on the surface temperature data (T2m field in ERA5). We split the dataset into individual samples, where each sample includes 2–4 days of surface temperature data (normalized to have unit variance) as well as the latitude and longitude from 60–90 randomly selected reporting stations. We also include the time of the first included timestep. We encode the coordinates by using the sine of the latitude and the sine and cosine of the longitude such that we preserve the periodicity. Similarly, we encode the time of year and time of day using the sine and cosine of the position along the 24 hour and 365 day cycles. We include all this information in a JSON format as follows[1]:

```
{'description':{'coords':[[1,-.32,.95] ... [.96,.61,.79]], 'start':[0,1,-.026,-1]},
'data':[[-2.6,-2.6 ... -3.2,-3.1,-3]]}
```

The `coords`, `start`, and `data` correspond to the reporting station coordinates, the time of the first sample, and the normalized temperature data, each reported separately per station and then concatenated in the data list. In this way, the model needs to parse both the textual aspects of the sample (e.g., where the commas appear to separate different parts of the data) as well as the numerical values. Furthermore, as is often the case with JSON-formatted data, the data does not have a causal format. We therefore train the language models using an MLM approach instead of the more common AR approach.

We evaluate the performance of the different numerical encodings on the task of predicting the next temperature timestep for all reporting stations simultaneously in a held out test set. We do so by masking the tokens (and numbers, if applicable) of all the data associated with the final timestep. Because the temperature data is provided separately per station, the masks are scattered throughout the input data and are not all simply at the end of the sample. Table 2 shows the results of this experiment in terms of compute: XVAL provides the best performance while taking considerably less compute time. Fig. 3 and Table 3 summarize

---

[1]For demonstration purposes, we show a few digits per number, but for both scientific datasets, all numbers are floating point numbers. For the text-based encodings, this text string is then processed according to the procedure described above.

Table 2: Performance (in MSE) and runtime of the different encodings on predicting the temperature for the next time step. "Equal Samples" columns refer to all models being trained for 500k iterations. Training was performed on 4 Nvidia H100 GPUs using Pytorch Distributed Data Parallelism.

| | Equal Samples | | Equal Tokens | | Equal Runtime | |
|---|---|---|---|---|---|---|
| Method | Loss | Runtime | Loss | Runtime | Loss | Runtime |
| P10 | 73 | 2d 22h | 73 | 2d 22h | 73 | 2d 22h |
| P1000 | 20 | 2d 2h | 23 | 3d 10h | 21 | 2d 22h |
| B1999 | 20 | 20h | 19 | 2d 23h | 19 | 2d 22h |
| FP15 | 2.14 | 19h | 1.76 | 3d 12h | 1.85 | 2d 22h |
| xVal | **1.75** | **9h** | **1.62** | **1d 15h** | **1.51** | **2d 22h** |

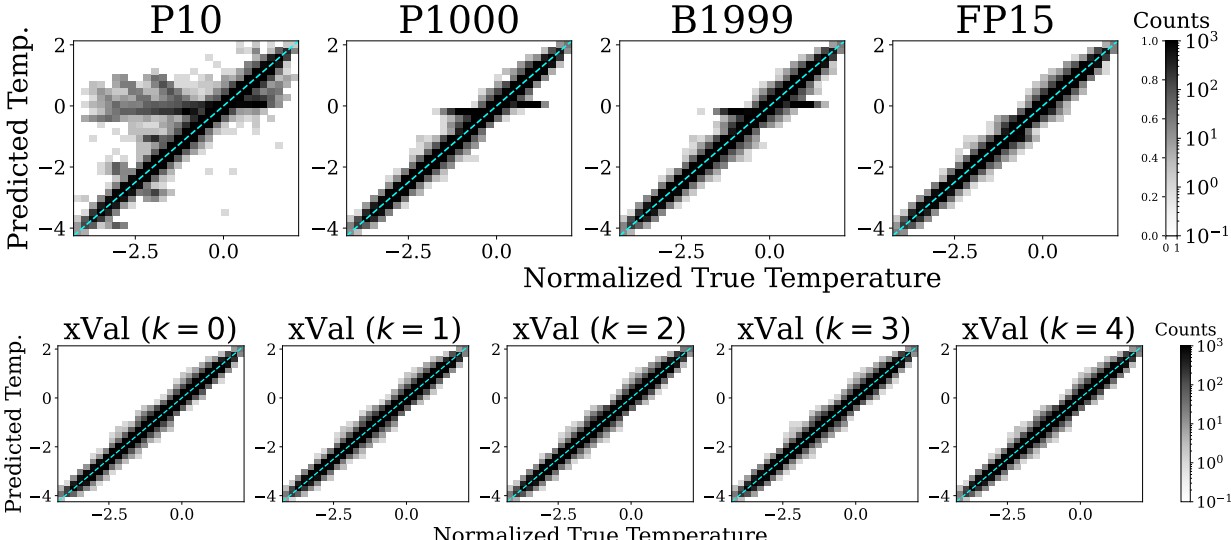

Figure 3: Performance of the encoding schemes in predicting the temperature of the next timestep for each reporting station in the ERA5 dataset. Mean Squared Error (MSE) values are reported in Table 3.

the performance of the various encodings on a held-out test dataset. Three indepdendent trials are performed for each encoding, and the median values across the three trials for each true data point are plotted.

This task also exemplifies one of the shortcomings of text-based encoding schemes: they can take advantage of spurious correlations in the data. In this case, P10, P1000 and B1999 have a tendency to predict normalized temperature $\pm 0.1$, which manifest as extended horizontal protrusions in Fig. 3.

These horizontal protrusions are due to the over-abundance of certain numbers in the dataset, as illustrated in Fig. 4. Individually, `100` and `E-3` are the most common numbers and exponents in the dataset, but when combined, `100E-2` is much more frequent than `100E-3`. This explains why FP15, which encodes the digits and exponents as one token, does not get confused in this case. It also implies that the model has failed to learn the correct joint distribution of the numbers. In these cases, because of the tokenization scheme, the length of the tokenized samples are very long, averaging around 8,000 and 5,000 tokens respectively for P1000 and P10 (compared to 1,800 tokens for FP15 and xVal). The poor performance in these models might therefore be due to the the challenges of modelling long-range interactions (Qin et al., 2023).

For more details on the performance of the different encodings, as well as comparison with some non-transformer baselines, see Appendix D.1. In Appendix D.3 we look at the performance of a BPE tokenizer on this task and demonstrate how LLMs can exploit the tokenized length of the number. In Appendix D.1.3 we train fine-tune these models on a simple binary classification task and compare their performance.

Table 3: Mean Squared Error (MSE) of the various encodings when predicting next-timestep temperature $(t+1)$ across all reporting stations simultaneously on the ERA5 dataset. The P10 encoding scheme has a very large variance between the three trials. Of the baselines, FP15 has the best performance, but the xVAL encodings are consistent with or better than any of the baselines.

| Encoding | ERA5 $t+1$ MSE |
|---|---|
| P10 | $(1.73 \pm 3.2 \times 10^5) \times 10^{-1}$ |
| P1000 | $(5.20 \pm 0.04) \times 10^{-2}$ |
| B1999 | $(5.27 \pm 0.10) \times 10^{-2}$ |
| FP15 | $(4.54 \pm 0.06) \times 10^{-3}$ |
| xVAL $(k=0)$ | $(4.29 \pm 0.30) \times 10^{-3}$ |
| xVAL $(k=1)$ | $(4.21 \pm 0.10) \times 10^{-3}$ |
| xVAL $(k=2)$ | $(4.18 \pm 0.03) \times 10^{-3}$ |
| xVAL $(k=3)$ | $(4.11 \pm 0.06) \times 10^{-3}$ |
| xVAL $(k=4)$ | $(4.18 \pm 0.08) \times 10^{-3}$ |

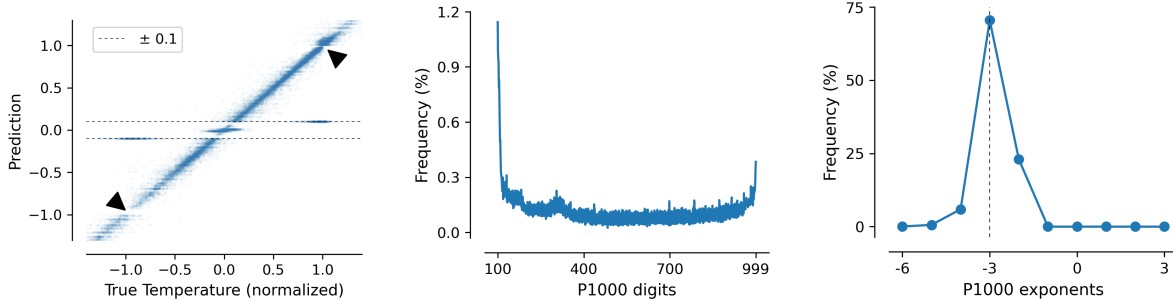

Figure 4: A failure mode of text based encoding scheme (left). Because of the distribution of the numbers in the training set (center and right), numbers close to $\pm 1$ (denoted by the black arrows) get misclassified as `100E-3`, i.e. 0.1, the combination of the most common digit and the most common exponent in the dataset.

## 3.2 Predicting planetary orbits

We then compare the performance of the various number encoding schemes on a simulated dataset of planetary orbits. We construct a dataset consisting of planetary motion simulations generated by the REBOUND N-body codebase (Rein, H. & Liu, S.-F., 2012) and integrated using IAS15 (Rein & Spiegel, 2015). The dataset consists of 1.25 million samples, split into 80%, 10%, 10% for training, validation, and test. Each sample consists of simulation parameters (mass and orbit properties of each planet and the simulation timestep size) as well as a sequence of $(x, y)$ positions for each planet, organized in a JSON format. The details of the simulation are provided in Appendix D.2. A typical sample in this dataset is given by:

```
{'description':{'planet0':{'m':2.38, 'a':2.96, 'e':1.73},
'planet1':{'m':1.35, 'a':1.31, 'e':1.5}, ... , 'stepsize':0.2},
'data':[[[2.60,-0.75],[0.81, 0.42]],[[2.63,-0.63],[0.70,0.60]]...]}
```

We pretrain the models using MLM and evaluate the models on the task of inferring the simulation parameters, specifically the simulation timestep $\Delta t$, and the semi-major axis, eccentricity and mass of the first planet $(a_0, e_0, m_0)$ by masking the appropriate locations. The models are also tasked with predicting the $(x, y)$ coordinates of each planet for each timestep in each simulation. The quantities $\Delta t$ and $a_0$ in the training corpus take values that are either discrete or leave gaps in the values they cover. This property makes these quantities particularly appropriate for testing interpolation generalization. Each tokenization is tested over three independent trials on a held-out test set, and the median values across the trials are used for the evaluation.

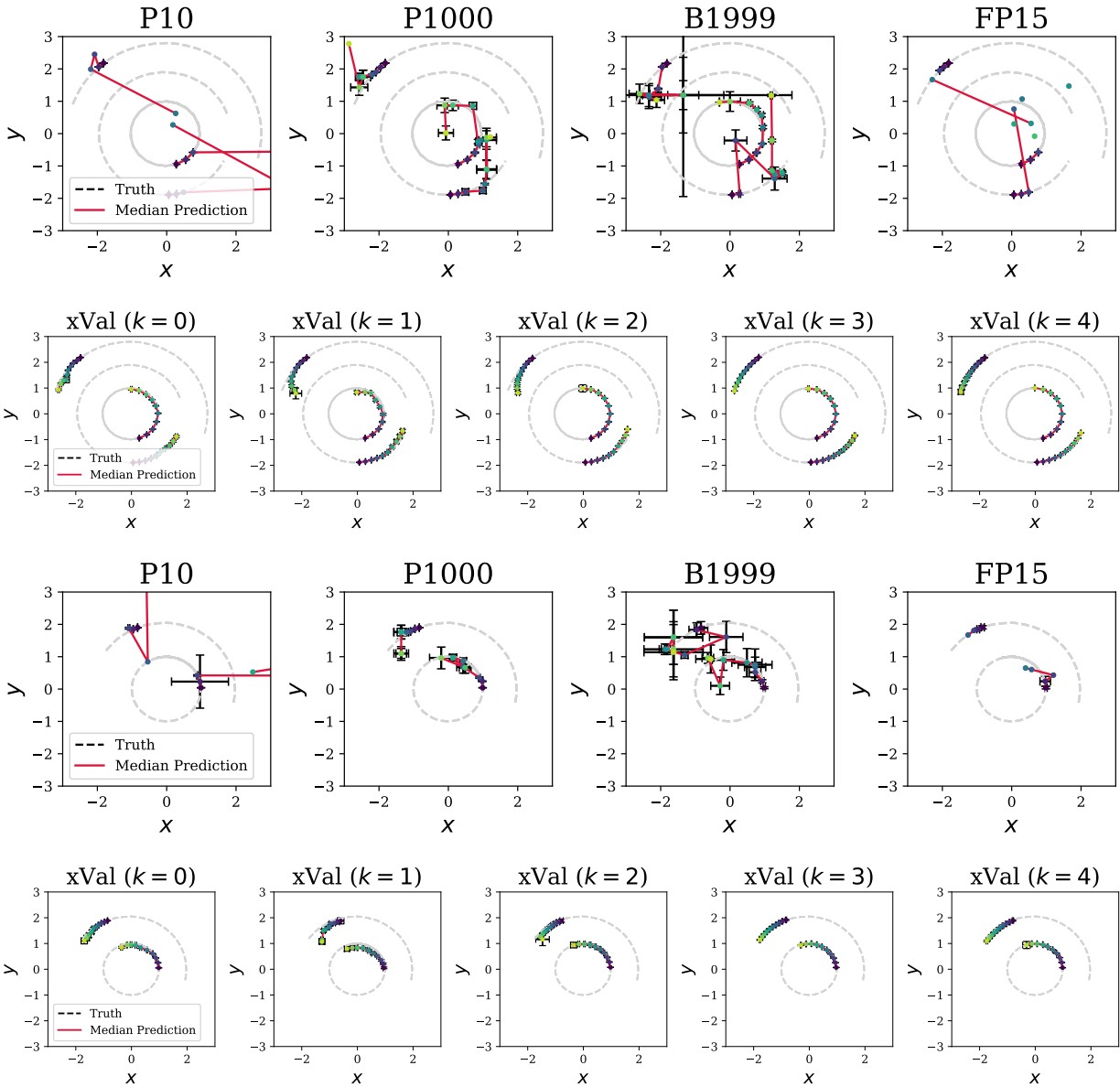

Figure 5: Performance of the encoding schemes predicting the 10 final timesteps of each planet for two simulated orbits. The prediction is not autoregressive: all 10 timesteps are predicted at the same time.

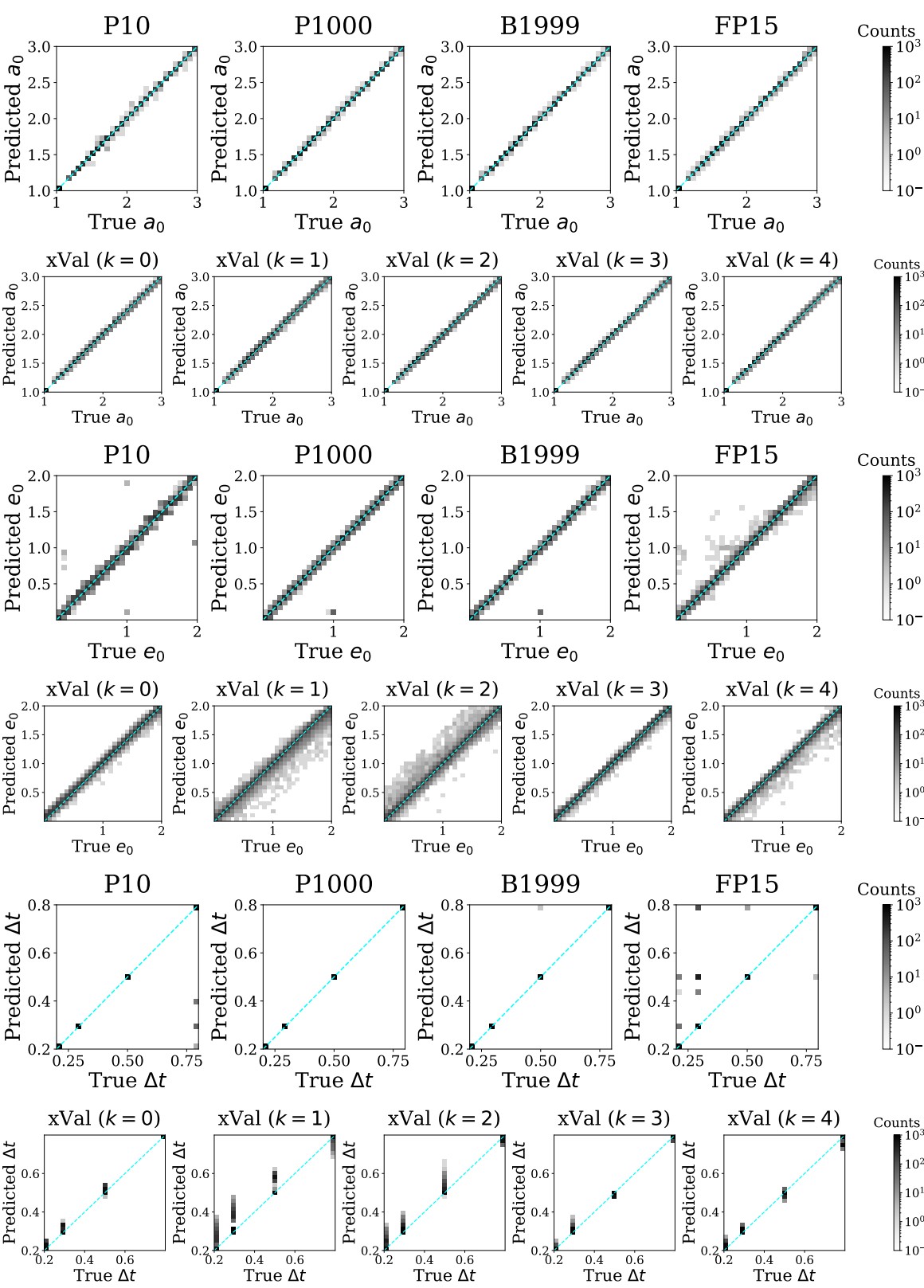

Figure 6: Performance of the encoding schemes predicting the semimajor axis $a_0$ and eccentricity $e_0$ of the first planet as well as the timestep sampling rate $\Delta t$ in the simulated dataset of planetary orbits. Mean Squared Error (MSE) values are reported in Table 4.

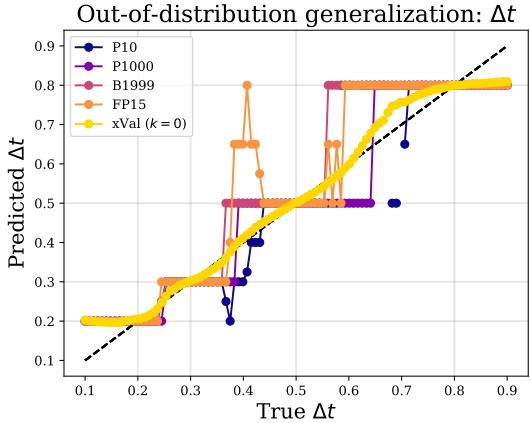
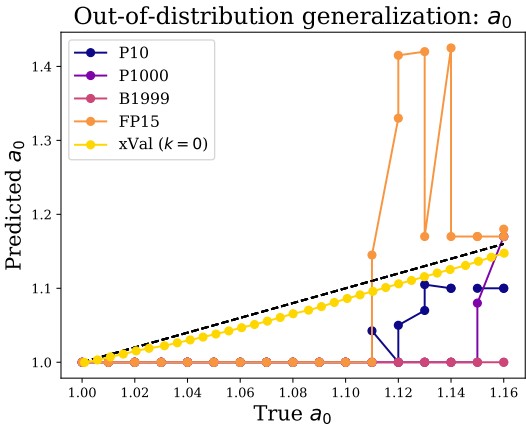

Figure 7: Out-of-distribution generalization properties of the different number encoding schemes. Left: Inferring $\Delta t$, which takes discrete values in the training set. Right: Inferring $a_0$, which is either 1 or $> 1.16$ in the training set. Because of the generation procedure, taking $a_0 \to 1.16$ here does not result in an in-train-distribution sample. Gray lines correspond to the discrete values of $\Delta t$ in the training dataset.

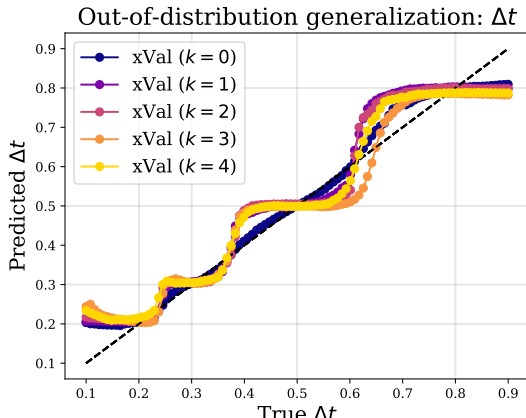
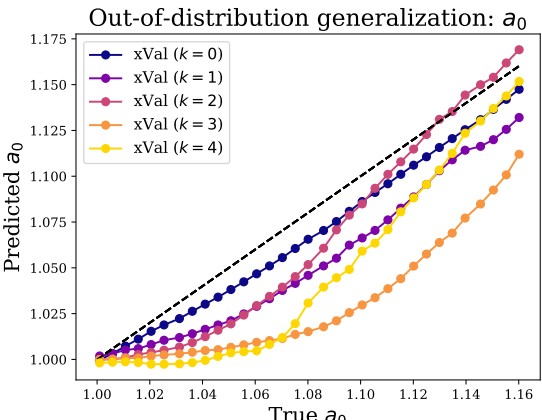

Figure 8: Out-of-distribution generalization properties of different configurations of xVal. In general, increasing the number of xVal tokens tends to degrate its out-of-distribution generalization performance. Left: Inferring $\Delta t$, which takes discrete values in the training set. Right: Inferring $a_0$, which is either 1 or $> 1.16$ in the training set. Because of the generation procedure, taking $a_0 \to 1.16$ here does not result in an in-train-distribution sample. Gray lines correspond to the discrete values of $\Delta t$ in the training dataset.

A summary of these experiments is presented in Table 4. Fig. 5 shows examples of predicting the final 10 timesteps simultaneously across the various tokenizations, and predictions of orbital parameters $a_0$, $e_0$, $\Delta t$ can be seen in Figs. 6, while $m_0$ is shown in Appendix A. Fig. 7 shows the out-of-distribution performance for the baseline methods compared with default xVal, while Fig. 8 compares xVal in various multi-scale configurations.

On the timeseries prediction task, none of the encoding schemes are able to consistently predict the planetary positions across the final ten timesteps simultaneously. When inferring overall orbital parameters, however, we generally see an overall inverse relationship between performance in- and out-of-distribution. For example, P10—the encoding with the fewest vocabulary elements—provides the worst in-distribution performance but performs much better on out-of-distribution tasks (see Fig. 7). This is an example of the bias/variance trade-off applied to the number of vocabulary elements. We see some examples of significant outliers across the

Table 4: Mean Squared Errors (MSEs) of the different numerical encodings on the planetary motion inference problem. Here, "OoD" ("Out-of-Distribution") indicates that the true value of a given parameter was part of a range not seen in the training corpus. ▶ indicates the lowest error for predicting each parameter.

| Method | Semi-major axis ($a_0$) | Timestep ($\Delta t$) | Eccentricity ($e_0$) |
|---|---|---|---|
| P10 | $(1.45 \pm 9.5) \times 10^{-1}$ | $(2.95 \pm 1.9) \times 10^{-2}$ | $(3.31 \pm 3.4) \times 10^{-2}$ |
| P1000 | ▶ $(4.00 \pm 0.0) \times 10^{-6}$ | ▶ $0 \pm 0$ | $(5.15 \pm 2.4) \times 10^{-3}$ |
| B1999 | $(4.00 \pm 4.6) \times 10^{-6}$ | $(1.80 \pm 236) \times 10^{-5}$ | $(3.45 \pm 2.6) \times 10^{-3}$ |
| FP15 | $(7.00 \pm 6.2) \times 10^{-6}$ | $(4.95 \pm 38) \times 10^{-3}$ | $(2.62 \pm 11) \times 10^{-3}$ |
| xVal ($k = 0$) | $(5.10 \pm 8.5) \times 10^{-5}$ | $(2.32 \pm 30) \times 10^{-4}$ | $(1.37 \pm 2.3) \times 10^{-3}$ |
| xVal ($k = 1$) | $(1.34 \pm 6.9) \times 10^{-4}$ | $(2.04 \pm 2.3) \times 10^{-3}$ | $(7.37 \pm 3.7) \times 10^{-3}$ |
| xVal ($k = 2$) | $(1.30 \pm 6.7) \times 10^{-4}$ | $(5.59 \pm 11) \times 10^{-4}$ | $(7.91 \pm 5.4) \times 10^{-3}$ |
| xVal ($k = 3$) | $(3.10 \pm 2.3) \times 10^{-5}$ | $(1.04 \pm 2.1) \times 10^{-4}$ | ▶ $(9.77 \pm 6.2) \times 10^{-4}$ |
| xVal ($k = 4$) | $(3.90 \pm 1.0) \times 10^{-5}$ | $(3.69 \pm 4.9) \times 10^{-4}$ | $(2.69 \pm 1.3) \times 10^{-3}$ |

| Method | OoD $a_0$ | OoD $\Delta t$ | Planetary Mass ($m_0$) |
|---|---|---|---|
| P10 | $(4.07 \pm 1.3) \times 10^{-3}$ | $(6.31 \pm 1.95) \times 10^{-3}$ | $(5.97 \pm 5.5) \times 10^{-1}$ |
| P1000 | $(6.99 \pm 20) \times 10^{-3}$ | $(4.69 \pm 0.34) \times 10^{-3}$ | $(3.33 \pm 0.028) \times 10^{-1}$ |
| B1999 | $(9.07 \pm 0.57) \times 10^{-3}$ | $(8.20 \pm 0.80) \times 10^{-3}$ | ▶ $(3.28 \pm 1.4) \times 10^{-1}$ |
| FP15 | $(1.18 \pm 28) \times 10^{-2}$ | $(1.04 \pm 0.41) \times 10^{-2}$ | $(1.19 \pm 0.35) \times 10^{0}$ |
| xVal ($k = 0$) | ▶ $(1.6 \pm 0.6) \times 10^{-4}$ | ▶ $(1.25 \pm 0.11) \times 10^{-3}$ | $(1.33 \pm 0.00035) \times 10^{0}$ |
| xVal ($k = 1$) | $(7.7 \pm 2.2) \times 10^{-4}$ | $(3.31 \pm 0.59) \times 10^{-3}$ | $(1.31 \pm 0.0011) \times 10^{0}$ |
| xVal ($k = 2$) | $(3.8 \pm 7.4) \times 10^{-4}$ | $(3.52 \pm 0.26) \times 10^{-3}$ | $(1.31 \pm 0.0060) \times 10^{0}$ |
| xVal ($k = 3$) | $(2.92 \pm 0.89) \times 10^{-3}$ | $(3.31 \pm 0.17) \times 10^{-3}$ | $(1.32 \pm 0.0022) \times 10^{0}$ |
| xVal ($k = 4$) | $(1.36 \pm 0.96) \times 10^{-3}$ | $(3.08 \pm 0.38) \times 10^{-3}$ | $(1.33 \pm 0.0018) \times 10^{0}$ |

four baseline encoding schemes, particularly in the predictions of $e_0$ and $\Delta t$ (Fig. 6). None of the baseline encodings are able to successfully predict $m_0$ (see Appendix A).

In comparison, xVal is able to correctly and accurately predict the final 10 timesteps of each planet's orbit, with particularly strong results for $k > 2$ (see Fig. 5). When inferring orbital parameters, xVal provides the best out-of-distribution performance while staying very competitive in-distribution. The out-of-distribution performance of the various baselines versus default xVal in Fig. 7. Here we see that the text-based encodings, with the exception of P10, simply do not predict any number that they did not explicitly see for this parameter in the training corpus. As expected from a function that is continuous by construction, xVal continuously interpolates between the values seen in the training set and offers much better performance. However, as seen in Fig. 8, increasing the number of xVal encoding tokens tends to degrade out-of-distribution generalization performance, even as it tends to improve in-distribution timeseries prediction performance.

Figure 7 shows that the predictions coming from the text-based encodings can be discontinuous when evaluated out-of-distribution. This discontinuity has two potential sources: the discontinuous nature of the number embeddings and the argmax that is taken over the logits during inference. Since the encodings of the number tokens in text-based encodings have been shown to form continuous-looking structures (see Sec. D.5 and Power et al. (2022); d'Ascoli et al. (2022)), it is possible that the discontinuity is only a side effect of the argmax and that the logits themselves vary more smoothly (see App. **??**).

### 3.3 Results summary

It is evident that embedding the magnitude of numbers directly, as in xVal, leads to a different inductive bias than treating numbers as tokenized text. This can be clearly seen in the varying performance of these language models in different tasks. When predicting the next timestep in the temperature dataset as well as the final 10 timesteps of each planet in the planetary dataset, xVal provides by far the best results. On

the other hand, in the mass prediction on the planetary task, none of the numerical encoding strategies are able to accurately learn the appropriate relationship.

Where xVAL particularly excels is in out-of-distribution performance, while the text-based encoding schemes fail to interpolate properly. The best interpolation for the text-based encodings is given by the vocabulary-sparse P10, which performs poorly on the in-distribution tasks. The extra encoding length of P10 also makes it prohibitively expensive to deploy, as shown in Table 2. The baseline encodings generally show the best in-distribution performance, but they have poor interpolation properties and expensive embedding costs. Overall, xVAL provides the best mix of in-distribution and out-of-distribution performance. Moreover, it is the most computationally efficient of the encoding schemes we considered.

**Failure modes.** There are a number of ways that number inference via a large language model can fail. The language model can predict a non-numeric token in the place of the number, leading to an invalid prediction. These are denoted in the percentages in brackets in Table 4, shown only when the percentage exceeded 0.01%. This failure mode is uncommon and becomes less frequent the more the model is trained. Another failure mode is when the model exploits spurious correlations. For example, the model can learn the distribution of the digits, as discussed in the example of temperature dataset, or the length of the encoding (see Appendix D.3).

A model can also fail to learn the correct distribution. In the planetary orbits example, learning the mass of the planet is the most challenging task – all encodings struggle with this. In this task, xVAL performs uncharacteristically poorly. We suspect that this is due to the high uncertainty in estimating the mass and that a multi-modal distribution such as the categorical distribution learned by traditional LLMs would perform better. This can be seen in Fig. 9. While each method performs poorly when considering the MSE of the prediction, the multi-modal predictions of the baseline models would be a better starting point for capturing an uncertain distribution. We therefore suspect that generalizing the xVAL number head such that it fits a mixture of Gaussian instead of predicting a scalar for each number would improve this performance.

## 4 Discussion

In this work, we introduced xVAL, a continuous number encoding that makes transformer-based models end-to-end continuous when considered as a function mapping the numerical values of the input to those of the output. We demonstrated that even though xVAL is more token-efficient and has a minimal vocabulary footprint, it excels in tasks on numerical datasets and leads to superior performance, especially when evaluated on out-of-distribution samples and time series predictions. Because of the fundamentally different treatment of numbers across these cases, xVAL and text-based encodings lead to different inductive biases, making the choice of the best encoding method on a given dataset highly dependent on the problem under consideration.

**Future directions.** As we have seen, using the xVAL encoding scheme renders the model not just continuous, but also differentiable as a function of the numbers it predicts. This enables the model to incorporate not just an MSE loss, but other statistical learning schemes. For example, we can add a Gaussian Mixture Model or any other differentiable loss and train the model to optimize this objective. This holds the promise to improve the experiments in which xVAL underperformed in this paper.

xVAL, combined with our proposed number-inference paradigm, makes language model architectures generally more suitable for processing scientific datasets. LLMs have become increasingly integrated in many scientific workflows today, enabling researchers to parse scientific language in sophisticated ways. However, their usefulness for analyzing numerically-heavy data is currently limited. Crafting specialized language models that have a better understanding of numerics has the potential to greatly increase their usefulness in scientific analysis and discovery.

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

# A    Predicting Planetary Mass

None of the encoding schemes are able to reliably and accurately predict the planetary mass of the first planet in each orbital simulation, as shown in Fig. 9. These plots also illustrate the typical failure mode of XVAL, which is trained using MSE loss: predicting the average value of the training data.

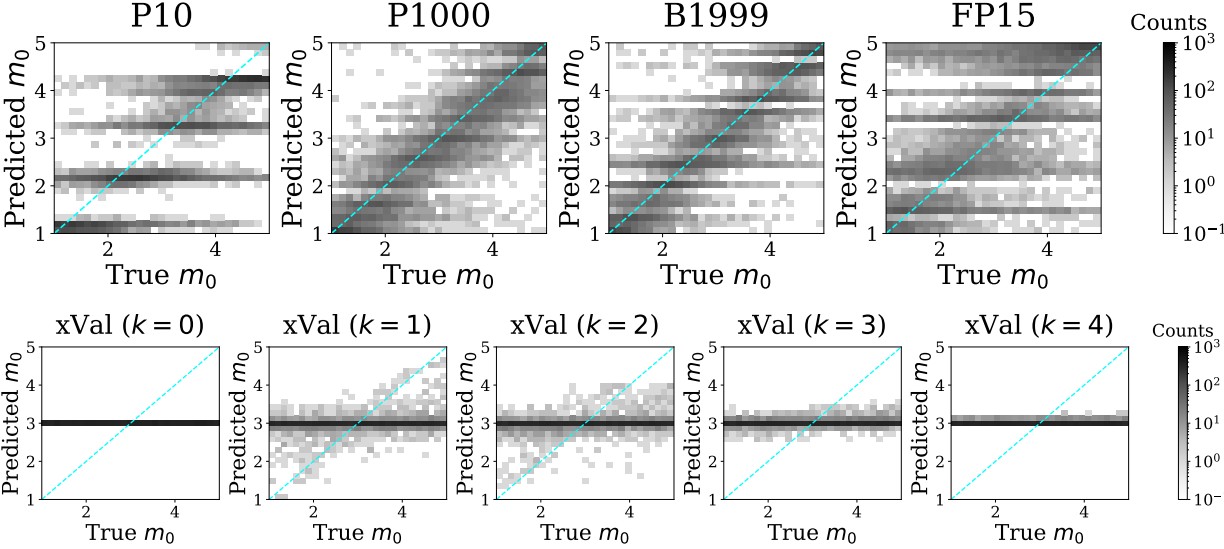

Figure 9: Performance of the encoding schemes predicting the mass $m_0$ of the first planet in the simulated dataset of planetary orbits. Mean Squared Error (MSE) values are reported in Table 4.

# B    Learning Arithmetic

Simple arithmetic problems have acted as a test bed for probing the mathematical reasoning abilities of language models (Dziri et al., 2023). In this section, we investigate the effect of the number encoding scheme on the ability of language models to perform multi-digit multiplications as well as multi-operand mathematical operations. Multi-digit multiplication is a notably challenging task for even the largest LLMs (Borji, 2023). Dziri et al. (2023) show that GPT-4 achieves only 59% zero-shot accuracy on three-digit multiplication problems, while its accuracy for four- and five-digit multiplication drops to 4% and 0%, respectively.

Table 5 reports the $R^2$ scores for multi-digit multiplication problems on several language models designed to handle numerical values. All number encodings generally perform well on this task. However, we find that some encoding schemes (P10 and FP15) show a tendency to yield a small percentage of highly erroneous predictions in some contexts, thereby reducing the $R^2$ score, while XVAL does not produce such outliers.

For a more challenging arithmetic task, we designed a dataset of multi-operand mathematical operations. We used random binary trees combining a fixed number of operands (2, 3, or 4) using the binary operators of addition, subtraction, and multiplication. build a dataset in which each sample is an arithmetic statement such as `((1.32 * 32.1) + (1.42-8.20)) = 35.592`. We then processed the samples according to the processing requirements of each number-encoding scheme. The task is evaluation of the expression on the left-hand side of the equation, implemented as a mask completion, where the right-hand-side number is masked. Table 6 shows the adjusted $R^2$ scores results on this task. XVAL performs remarkably well on this task.

Table 5: Number multiplication experiment. Adjusted $R^2$ scores calculated between predictions and true values for the different encodings on various arithmetic datasets. (Higher is better; $R^2 = 1$ is the theoretical maximum.)

| Encoding | 3-digit | 4-digit | 5-digit |
|---|---|---|---|
| P10 | 0.9989 | 0.6071 | 0.9439 |
| P1000 | 0.9997 | 0.9783 | 0.9991 |
| B1999 | 0.9998 | 0.9984 | 0.9997 |
| FP15 | 0.7119 | 0.9959 | 0.9980 |
| xVAL | 0.9986 | 0.9975 | 0.9958 |

Table 6: Arithmetic evaluation task of random binary trees combining different numbers of operands with addition, subtraction, and multiplication. $R^2$ measured between true expression value and transformer prediction.

| Encoding | 2 operands | 3 operands | 4 operands |
|---|---|---|---|
| P10 | 0.998 | 0.996 | 0.992 |
| P1000 | 0.991 | 0.990 | 0.991 |
| FP15 | 0.993 | 0.981 | 0.935 |
| xVAL | 0.99998 | 0.99994 | 0.99998 |

Arithmetic experiments alone are not sufficient for fully evaluating the mathematical abilities of language models. The samples in these datasets are often short sequences and the underlying data manifold is low-dimensional. These problems therefore do not push the boundary of what is computationally possible with LLMs.

## C    Architecture details

In our experiments, all the language models, regardless of encoding, adopt the main features of GPT-2 (Radford et al., 2019). That is, we use absolute position encoding and the transformer blocks have layer norms prior to the attention module and the MLP (i.e., after each residual connection). We also set the width of the MLP hidden layer equal to 4 times the width of the embedding. We deviate from GPT-2 in that we initialize all the weights of the transformer blocks with a normal distribution with standard deviation given by $(2 \times \text{fan-in} \times \text{num-layers})^{-1/2}$. The dependence of the standard deviation on the number of transformer blocks is to counteract the effect of having a series of residual connections. We also do not use any biases in the trunk of the transformer. As is standard, after the transformer blocks, we have a token-head, comprised of a single linear layer, which maps the latent embedding of each token into a distribution over the vocabulary. As in GPT-2, we tie this weight to that of the embedding matrix which maps the tokens of the input to the embedding space.

For the LLMs using xVAL encoding and an MSE number-head in addition to the token head, we promote both heads (number and token) to be MLPs with one hidden layer of width equal to the embedding dimension. This was to allow the two different prediction types (the number and the distribution over the vocabulary) to be processed separately before the final prediction. In particular, we explore the possibility of having biases for the number-head and not in the token-head in Sec. D.4.

For all of our training runs, we use a cosine learning-rate schedule with warm-up. The scheduler is adjusted such that it reaches the minimum learning rate at the end of the training run.

# D Further experimental details

In our experiments, the hyperparameters for Learning Rate via a grid search coarse grid search in log space with spacing given by a factor of 2. We chose the best performance on a validation set and reported the results on an unseen test set. We have added this description to the supplementary materials section. The result of the search can be seen in tables below.

## D.1 Temperature forecasting

### D.1.1 Experiment details

**Dataset details.** The ERA5 dataset (Hersbach et al., 2020) is a high-resolution, state-of-the-art global atmospheric reanalysis product provided by the European Centre for Medium-Range Weather Forecasts (ECMWF). It is the fifth generation of ECMWF atmospheric reanalyses and represents the latest advancement in the ERA (ECMWF Re-Analysis) project. The dataset covers the period from 1979 to near-real-time and is updated regularly.

In our experiment, we take only the surface temperature of the dataset (field T2m) sampled at 8 hour intervals. For each sample, we randomly choose 60–90 of the ∼1-million spatial grid points of the dataset, and include 8–16 temperature time points at 8-hour intervals (corresponding to 2–4 days), starting from a random time. We generate 1.25 million examples in this way and split it into 1 million train, 125 thousand validation and test set samples.

```
{'description':{'coords':[[1,-.32,.95] ... [.96,.61,.79]],
'start':[0,1,-.026,-1]}, 'data':[[-2.6,-2.6 ... -3.2,-3.1,-3]]}
```

The samples are individually preprocessed such that the temperature range across all samples has mean zero and standard deviation equal to 1. We also include the latitude and longitude information. To respect the periodicity of this information, we provide the sine of the latitude and the sine and cosine of the longitude. Furthermore, we specify the starting time for each sample as the day of year and time of day. Again to respect the periodicity of these quantities, we provide the sine and cosine of the phase of these quantities.

**Architecture design hyperparameters.** For all experiments done with this dataset, we use transformers with 6 transformer blocks, each with 6 heads and each head having width 128, resulting in a embedding width of 768 (43.5M parameters).

**Training hyperparameters.** For the equal samples training runs, we train each model for 500k iterations with batch size equal to 64 samples. For the equal tokens runs, we increase the number of iterations proportionately such that the total number of tokens seen is equal. This implies: 500k samples for P10, 820k for P1000, 1.2M for B1999, and 2.3M for FP15 and xVAL. Since there is non-numeric data in the samples, the ratio of the length of the equal tokens is slightly different from the ratio of the length of each encoding scheme's tokenization length for numbers. The other hyperparameters in this task are given in Table 7.

Table 7: Training hyperparameters for the different encodings on the Temperature Forecast dataset.

| Encoding | Learning Rate | Minimum LR | Warmup | Max Context Length |
|---|---|---|---|---|
| P10 | $2.5 \times 10^{-5}$ | $2.5 \times 10^{-6}$ | 2000 | 8222 |
| B1999 | $10^{-4}$ | $10^{-5}$ | 2000 | 1251 |
| P1000 | $10^{-4}$ | $10^{-5}$ | 2000 | 5010 |
| FP15 | $10^{-4}$ | $10^{-5}$ | 2000 | 1798 |
| xVAL | $2 \times 10^{-4}$ | $2 \times 10^{-5}$ | 2000 | 1798 |

### D.1.2 Non-transformer baselines

To understand this task better, we trained a number of non-transformer baselines for comparison. These models are reported just for comparison and by no means represent the best possible non-transformer based baslines.

First, we looked at the performance of an MLP model when trained in a supervised way to predict the next time step (All stations). To deal with the varying number of locations and varying number of time-steps, we simply keep the number of locations/time-steps that is the minimum across all samples (60 locations and 8 time-steps.) We then looked at the possibility of temperature forecast based on a single reporting station (Single Station). And then on this single-station dataset, we looked at the performance on the temperature data alone (Single Station - temp all), temperature data + station coordinate (Single Station - temp + coord), and temperature data + first time step time of year (Single Station - temp + ToY).

The MLPs acting on single stations have 3 hidden layers of width 256. The MLP looking at 60 stations simultaneously is larger to validate that the poorer performance is not because of limited network size. We tried width from 256–8192 and up to 5 layers and the results remain similar.

Table 8: Temperature forecast MLP baselines

| Method | MSE Loss (C) |
|---|---|
| All Stations | 2.31 |
| Single Station | 1.57 |
| Single Station - Temp only | 1.79 |
| Single Station - Temp + Coord | 1.65 |
| Single Station - Temp + ToY | 1.74 |

The results of these tests can be seen in Table 8. We see that for good performance, it is important for the model to have access to both the time of year as well as the coordinate of the reporting station. However, providing the information for multiple reporting stations at once makes the performance worse.

This implies that for the transformer model to be able to predict the temperature with MSE less than 1.7, it needs to properly parse all this information that is scattered across the different parts of the input string. xVal was the only model to achieve MSE below that of the MLP model (Table 2) meaning that it has likely learned to leverage the temperature of other reporting stations as well.

### D.1.3 Comparison of fine-tuning behavior

In this section, we explore the fine-tuning behavior of the different encoding schemes in a simplified setting. In this problem, we fine-tune a downstream model to predict whether or not the location of the first reporting station in the sample is located on the ocean. As this is a binary classification task, we train logistic regression on the final embedding of the transformer (the output of the last transformer block). We use 500 training samples for this task. While this problem is in principle solvable by looking at the latitude and longitude of the reporting station which is included in the data, 500 samples is not enough to learn this map. Therefore, the model needs to leverage other information in the temperature patterns to make this prediction. Table. 9 reports the performance of these models. We report the ROC AUC as a more balanced metric, since the distribution of land vs ocean on earth is not symmetric.

Table 9: Performance of the different number encoding schemes when fine-tuned on the binary task of predicting whether the first reporting station is on the ocean or on land.

| Method | ROC AUC |
|---|---|
| P10 | 0.580 |
| FP15 | 0.600 |
| xVal | 0.62 |

## D.2 Planetary motion

**Dataset details.** In this dataset we use the REBOUND N-body simulation codebase (Rein, H. & Liu, S.-F., 2012) and IAS15 integrator (Rein & Spiegel, 2015) to generate a number of planetary systems (with a central mass $m_\odot \equiv 1$) and follow their orbits for a number of time points. Each planetary property is drawn from a uniform prior: the number of planets $n \in [2, 4]$, mass $m/m_\odot \in [10^{-5}, 5 \cdot 10^{-5}]$, semimajor axis equally spaced for the planets between 1 and $a_f \in [1.5, 3]$ (i.e. if 3 planets and $a_f = 1.8$ then $a_1 = 1$, $a_2 = 1.4$ and $a_3 = 1.8$), eccentricity $e \in [0, 0.1]$, and starting angle in the $(x, y)$ plane equal to zero for 30% of the samples and uniform $\theta \in [-\pi/6, \pi/6]$ for the remainder. These choices are made such that when generating the large number of samples required for training, we do not come across instabilities or collisions. Finally, we use an integration step-size sampled uniformly from $\{0.2, 0.3, 0.5, 0.8\}$.

We generate 1.25 million examples in this way and split it into 1 million train, 125 thousand validation and test set samples. We normalize the masses such that they take value between 1 and 5 and the eccentricities such that they are between 0 and 2. We then construct a JSON format sample including all of this information. A generic sample is given in this example.

```
{'description':{'planet0':{'m':2.38, 'a':2.96, 'e':1.73},
'planet1':{'m':1.35, 'a':2.96, 'e':1.73}, ... , 'stepsize':0.2},
'data':[[[2.60,-0.75],[0.81, 0.42]],[[2.63,-0.63],[0.70,0.60]]...]}
```

**Architecture design hyperparameters.** Similar to the Temperature Forecasting dataset, for all experiments, we use transformers with 6 transformer blocks, each with 6 heads and each head having width 128, resulting in a embedding width of 768 (43.5M parameters).

**Training hyperparameters.** We train each model for 500k iterations with batch size equal to 64 samples. The hyperparameters in this task are given in Table 10.

Table 10: Training hyperparameters for the different encodings on the Planetary Motion dataset.

| Encoding | Learning Rate | Minimum LR | Warmup | Max Context Length |
|---|---|---|---|---|
| P10 | $10^{-4}$ | $10^{-5}$ | 2000 | 2707 |
| B1999 | $10^{-4}$ | $10^{-5}$ | 2000 | 1251 |
| P1000 | $10^{-4}$ | $10^{-5}$ | 2000 | 1736 |
| FP15 | $10^{-4}$ | $10^{-5}$ | 2000 | 767 |
| xVal | $2.5 \times 10^{-5}$ | $2.5 \times 10^{-6}$ | 2000 | 767 |

## D.3 Erratic behavior of number encodings of unfixed length

In many JSON formatted datasets, the data does not follow a causal pattern, i.e. earlier entries might depend logically on latter entries. This is also the case for our JSON formatted samples. Because of this we used Masked Language Modeling (MLM) for pretraining our models. In the context of MLM, number encodings that lead to encoding lengths that vary based on the number can prove troublesome both during training and during testing. During train time, the length of the encoding acts as a cue to help the model figure what the number is. This is an example of spurious correlations that LLMs are known to exploit (Tu et al., 2020; Liu et al., 2022; Dziri et al., 2023). Similarly at test time, the length of the mask can bias the model toward predicting one number or another.

As a demonstration of this feature, we first preprocessed the Temperature Forecast dataset such that every number has only two significant figures, dropping the leading zeros for efficiency (e.g. $0.12 \rightarrow .12$).[2] We then used a tokenizer that included single and double digits as well as $\pm$, the decimal point and exponents ranging from (E-8 to E+2). In this dataset, positive and negative floats with magnitude between 0.1 and

---

[2]In the experiments of the Sec. 3, the numbers have three significant figures. Therefore the results of this section are not directly comparable to those of the main text.

1 (e.g. .23 and -.34) would have encoding lengths equal to 2 and 3 and positive and negative floats with magnitude between 0.01 and 0.1 (e.g. -.034 = 3.4E-2) would have encoding lengths 4 and 5. There are exceptions however. For example in this scheme 0.030=3E-2 has encoding length 2.

The results of this experiment can be seen in Fig. 10. We see that even though the model's overall performance is not great, it can tell with very high accuracy the number's sign, whether or not it has absolute value greater/less than 1, or greater/less than 0.1. This is due to the fact that the model is exploiting the correlation of the numbers with the length of the encoding. We verify this by highlighting in orange the cases where in the range between 0.01 and 0.1, the number has encoding length 2, that is it does not follow the general trend mentioned above. We see that the model believes that these numbers are greater than 0.1 (which as we saw generally had encoding length 2).

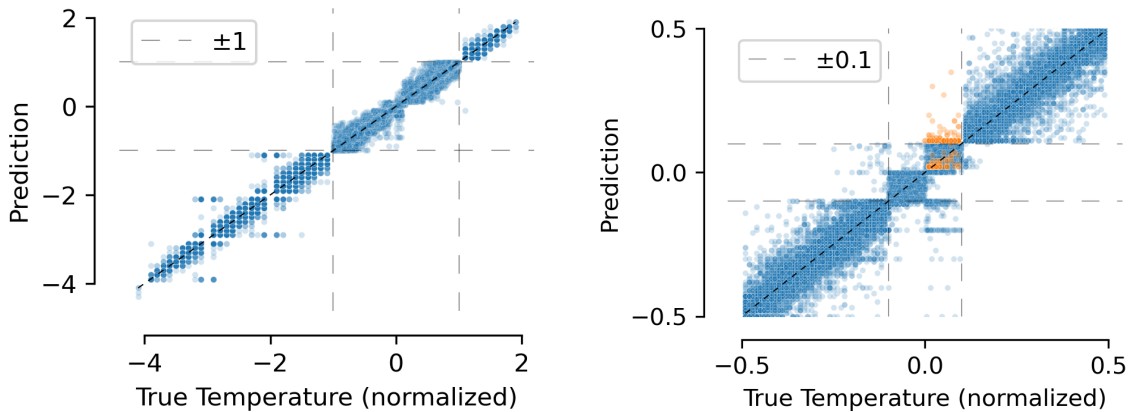

Figure 10: LLMs can exploit spurious correlations in the data. In this case, the model has learned the correlation between the number signs/values with the length of the encoding. Highlighted in orange are numbers between 0 and 0.1 that do not have the encoding length equal to 2..

### D.4 Architectural explorations

There are a number of engineering choices that we made regarding the architecture and hyperparameters of the transformer models trained with xVAL and the number head. Here, we explore the effect of these on the Temperature Forecast task. Because of the large exploration space and the high amount of compute required, we do the ablation tests on a shorter run, 100k iterations compared to 500k iterations of the main text. For this exploration, we first run all of the configurations with 4 different learning rates (2.5E-5, 5E-5, 1E-4, 2E-4). We then choose the best performing learning rate for each configuration and then run each configuration two more times with this learning rate. The result of this exploration is given in Table 11.

We summarize the various configurations that we run this experiments in and their effects as follows:

- Ratio of the final learning rate of the cosine scheduler to the initial learning rate (min-LR/LR). We found decreasing this ratio from 0.1 to 0.01 does not affect performance in this experiment. But we found that it does increase stability in longer runs.

- Turning off the layer norm prior to the MLP of the first transformer block (First Layer Norm = False). This change does not affect average performance. This is not surprising since the effect of the layer norm at this stage is simply to normalize the numbers and the numbers in this dataset are in the regime where the normalization discussed in Sec. 2.1 is linear.

- Turning off the layer norm prior to the MLPs of all transformer blocks (MLP Layer Norm = False) This change had a significant negative impact on the performance of the model.

Table 11: Ablation tests for the various design choices. Here Normal refers to min-LR/lr=0.1, Weight decay = 0.1 and MLM probability = 0.2, and the opposite dichotomy for the other choices.

| Configuration | Best Validation Loss | Learning Rate |
|---|---|---|
| Normal | $(6.8 \pm 0.2) \times 10^{-3}$ | 0.0002 |
| min-LR/LR = 0.01 | $(7.0 \pm 0.1) \times 10^{-3}$ | 0.0002 |
| First Layer Norm = False | $(6.8 \pm 0.5) \times 10^{-3}$ | 0.0002 |
| MLP Layer Norm = False | $(9.0 \pm 0.1) \times 10^{-3}$ | 0.0001 |
| MLM probability = 0.1 | $(8.2 \pm 0.6) \times 10^{-3}$ | 0.0002 |
| MLM probability = 0.3 | $(6.4 \pm 0.4) \times 10^{-3}$ | 0.0002 |
| Weight decay = 0.0001 | $(8.2 \pm 0.6) \times 10^{-3}$ | 0.0002 |
| Weight decay = 1 | $(5.3 \pm 0.3) \times 10^{-3}$ | 0.0002 |
| Trunk bias = True | $(6.2 \pm 0.4) \times 10^{-3}$ | 0.0002 |
| Num-head bias = False | $(6.9 \pm 0.1) \times 10^{-3}$ | 0.0002 |

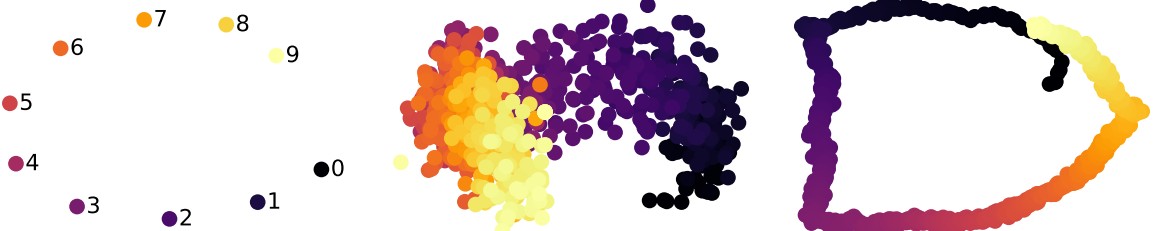

Figure 11: Two-dimensional PCA projection of the learned embeddings for mantissa tokens. (left) P10 encoding trained on the planet dataset; (center) P1000 encoding trained on the planet dataset; (right) P1000 encoding trained on the arithmetic dataset. Brighter colors denote higher number values.

- Changing the masking probability to 10% or 30% (default is 20%). Decreasing (resp. increasing) this probability lead to performance deterioration (resp. improvement) in this experiment. However, this seems to be dependent on the dataset as in other instances 30% seems to be too high for effective learning.

- Changing the weight decay to 0.0001 or 1 (default is 0.1). Increasing this value lead to the largest improvement. However, similar to the masking probability, this seems to be dataset dependent. The effect of increased weight decay can also depend on the length of the run.

- Including a bias in the modules of the transformer block (they are absent by default). Including this bias improved performance at the cost of increased variability.

- Turning off the bias in the number head (present by default). This change did not affect the performance significantly.

### D.5 Learned embeddings for text-based number encodings

Figure 11 shows the structure of number embeddings learned on different datasets for different encodings. For P10 the models learn rotary structure which is reminiscent of other works such as grokking (Power et al., 2022), and allows recovering relative numbers from inner products. It is also interesting to see how different datasets can lead to different learned encoding structures, for instance the arithmetic tasks seem to induce a more precise curve structure, while the planet data leads to more spread out embeddings, perhaps because the task is less sensitive to small perturbations of the numbers.

### D.6 Examining the logits

Fig. 12 shows an example of the logits of the P1000 encoding when predicting the step-size out-of-distribution. Here, the color lines denote the highest-value logits, with the other logits carrying negligible weight. The dashed gray lines denote the values of the step-size seen in the training set. We see that these lines are smooth in neither small or larger scales. We expect that this is a combination of the text-based number encodings' discrete embedding schemes together with the cross-entropy training paradigm that does not incorporate number distances into the loss.

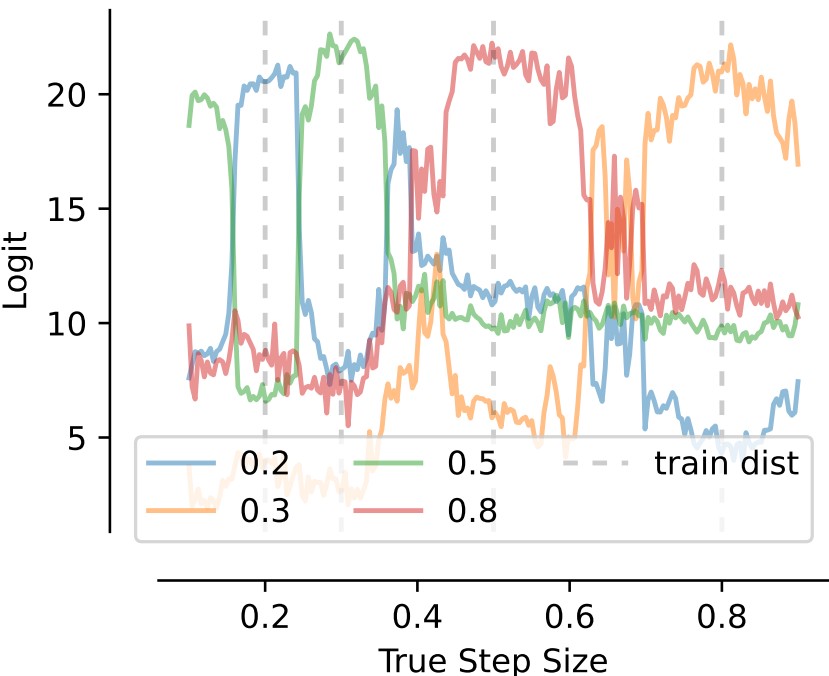

Figure 12: An example of the logits of model trained with P1000 encoding evaluated on the step-size prediction task.

### D.7 A number look-up experiment

In order to explore potential deleterious effects of using XVAL which encode the values of the numbers multiplicatively we set up a simple number lookup experiment. In this problem, we train the model on textual samples that demonstrate a dictionary lookup task:

```
"{d:1.53, e:-1.33, a:2.53, i:0.0232} e=-1.33"
```

In this experiment, we sample the number between -3 and 3 but we withhold a small region between 0.3 and 0.5 such that they do not appear in the dataset. As with our other experiments, we train a transformer model using self-supervised learning, in this case mask-filling with a masking probability of 30%.

In this experiment, we evaluate two metrics. The first is the Mean Square Error between the looked up number and the actual number. The second is the accuracy of reconstructing the number exactly as it appeared in the dictionary. Because our comparison textual number encodings have only 3 significant figures, we evaluate the accuracy of this lookup up to 3 significant figures and also provide the 2 significant figure accuracy for comparison. The results of the in-distribution and out of distribution evaluations are given in Tabs. 12 and 13. XVAL performs well on MSE but is unable to reconstruct the number exactly. The text encodings do better on accuracy but because of outliers, their performance on the MSE suffers. Furthermore, the text based encodings' performance drastically degrades on the held-out set whereas XVAL's does not degrade to the same extent.

**Note.** Because of the residual connections of the transformer, the information regarding the number value is principle present at readout time. Because of this, the network can in principle recover the value in context.

Table 12: Performance of the different encodings on the number lookup experiment. FP15 provides the best accuracy and xVal provides the best MSE.

| | | Accuracy | |
|---|---|---|---|
| Method | MSE | 3 Sig Figs | 2 Sig Figs |
| P10 | $(2.1 \pm 0.4) \times 10^{-2}$ | $91\% \pm 3\%$ | $94\% \pm 2\%$ |
| FP15 | $(7 \pm 0.7) \times 10^{-3}$ | $99.9\% \pm 0.1$ | $99.9\% \pm 0.1\%$ |
| xVal | $(3 \pm 0.5) \times 10^{-3}$ | $6\% \pm 1\%$ | $55\% \pm 5\%$ |

Table 13: Out of distribution performance of the different encodings on the number lookup experiment. Neither textual encoding schemes generalize to unseen values.

| | | Accuracy | |
|---|---|---|---|
| Method | MSE | 3 Sig Figs | 2 Sig Figs |
| P10 | $(3.8 \pm 0.3) \times 10^{-1}$ | $65.7\% \pm 2\%$ | $66.5\% \pm 1\%$ |
| FP15 | $(0.9 \pm 0.4) \times 10^{-1}$ | $34.5\% \pm 0.5$ | $34.5\% \pm 0.5\%$ |
| xVal | $(3.5 \pm 0.5) \times 10^{-3}$ | $1.5\% \pm 1\%$ | $12\% \pm 2\%$ |

