# OpenReview forum: "xVal: A Continuous Numerical Tokenization for Scientific Language Models"
_TMLR — Rejected by TMLR_

### Review · Reviewer_116U · 2025-02-03

**Summary Of Contributions:**

This paper pointed out that the encoding of numerical values is not continuous in latent space when using LLM for scientific datasets. The proposed method aims to achieve inference that is good at handling numerical values by using continuous embeddings with token efficiency.
In the case of using numerical values with existing LLM, the straightforward method is to encode them as text. In this case, we introduce not only the approach of preprocessing by adapting the format of the numbers, but also the approach of learning the embedding of the numbers. The proposed method emphasizes the continuity of the numbers and proposes a method of embedding them as a single variable. This can handle a wide range of numbers by applying appropriate scaling.
The proposed method has been verified on several datasets and has been shown to have superior performance in the numerical prediction task compared to existing tokenization methods.

**Audience:**

Yes

**Broader Impact Concerns:**

I don't have any immediate concerns about this research.

**Claims And Evidence:**

Yes

**Requested Changes:**

I have clarified whether requested changes or optional in the above section.

**Strengths And Weaknesses:**

Strengths:
- The idea of embedding based on numerical continuity is based on the important property that real numbers are a completely ordered set. This is a property that people intuitively understand and naturally use when extrapolating numerical continuity. The idea of the paper is clear, and it is expected to contribute to improving performance.
- The presentation of the paper is good and the content is easy to understand. In particular, Figure 1 shows the proposed method clearly and simply.
- Section 3.1: "This task also exemplifies one of the shortcomings of text-based encoding schemes: they can take advantage of spurious correlations in the data." The detailed analysis of this problem is highly appreciated as a discussion of the effectiveness of the proposed method.

Weaknesses (Requested to comment or change):
- The biggest weakness of the paper is that the tasks to be verified are limited to numerical prediction tasks. If the task is simply prediction, there is no need to use LLM. I hope that the evaluations will be done according to the concept of Scientific Language Models, combining scientific data with linguistic expression and description.
- This experiment is too numerical and deals with a well-developed problem as a continuous value prediction task. In such cases, the appropriateness of using a language model is questionable, and it would be desirable to consider scenarios that require language models more. In particular, it is expected that there are cases where textual encoding is more appropriate than numerical encoding. (This is especially true for cases where the specific number is important, such as the case of small integers).
- It is unclear what criteria are used to treat numbers as numbers when tokenizing actual strings. Not all digits in plain text are continuous numbers. For example, it is possible to assume that "1", "1.00", and "1E0" have different meanings. I hope there will be a discussion about the criteria and limits of tokenization.
- In scientific and technical calculations, we deal with numbers on a variety of scales. These are often described by exponents. In such cases, it is possible to assume that they can be encoded by splitting them into an exponent part and a mantissa part, but it would be better to have an explanation of why this is inappropriate. I assume that the same numerical encoding is applied to numbers with different meanings, and that this is not valid from the perspective of numerical continuity, which is the idea behind the proposed method. Is this correct?
- If you are dealing with a wide range of numerical values, applying scaling will cause the same problems as general numerical calculations, such as loss of trailing digits and elimination of significant digits. If you don't use scaling, it will be difficult to deal with the values at all range. Paradoxically, the solution to this problem may be to encode the values by splitting them into an exponent part and a mantissa part.
- Public Comment I think that the papers [2-5] presented in Alex Shtoff's comments are papers that should be cited in terms of encoding and handling numerical values. However, my review evaluates the paper from the perspective of numerical embedding for LLM, and I do not think that the novelty of the paper is lost by the fact that the mapping is an existing technology. Rather, I judged the weakness to be that the evaluation is inadequate as an LLM-required task.
- Section 2.1: "Continuous embeddings have been previously proposed for use in attention mechanism in the context of speech recognition (Chorowski et al., 2014)." Since the relationship between the cited paper and continuous embeddings was unclear, it would be better to discuss in more detail which methods have already been proposed and whether they are similar or different ideas.

Weaknesses and Comments (Optional):
- When dealing with numerical values, there are cases where the values should be treated as continuous values, and cases where the number of values is important (e.g., 4-dimensional space-time, 3rd order precision numerical calculations, etc.). It is possible to discuss whether numerical values that are continuous and numerical values that have meaning in the latter case should be treated in the same way. This is a comment intended to show that it is an overstatement to say that "all numerical values should be treated as continuous values". (This is not a weakness of the paper, but rather something that should be addressed in future research, including, for example, a survey of what numerical values appear in the text).
- Related to the previous point, it would be desirable to have a discussion about how to handle cases where the value of a number is significant, such as scientific constants (e.g., pi, e). In science, these numbers are generally given specific meanings, rather than being treated as just real numbers. I think this way of thinking suggests that it is not always the case that all numbers should be expressed in continuous form when processing scientific data in LLM.
- Adjusting the meaning of numbers using coefficients is equivalent to considering the unit system in a somewhat flexible understanding. For example, it would be desirable to discuss whether it is appropriate to treat kilo- and milli- in the same way, or whether they should be treated linguistically.
- This is not essential, but I am interested in the output of the number head corresponding to words that are not predicted to be of the [NUM] class (e.g., how do "desk", "chair", and "pen" correspond as numbers?) If there are any interesting trends, I would like to know about them.

---

> ### Author Response · Authors · 2025-02-22
>
> Dear Reviewer 116U,
>
> Thank you very much for your careful read of our paper and for your suggestions for improvement. We include our point-by-point replies below:
>
> - The biggest weakness of the paper is that the tasks to be verified are limited to numerical prediction tasks. If the task is simply prediction, there is no need to use LLM. I hope that the evaluations will be done according to the concept of Scientific Language Models, combining scientific data with linguistic expression and description. This experiment is too numerical and deals with a well-developed problem as a continuous value prediction task. In such cases, the appropriateness of using a language model is questionable, and it would be desirable to consider scenarios that require language models more. In particular, it is expected that there are cases where textual encoding is more appropriate than numerical encoding.
> 	- We like to make a distinction between Scientific *Language* Models vs. Scientific *Foundation* Models: the former are concerned with adapting LLMs for scientific reasoning tasks, while the latter need only serve as good pre-training baselines for fine-tuning on mostly-numerical scientific datasets. We focus on the latter case, meaning that we intend xVal to be used to process numerical scientific datasets with minimal natural language present, usually just in the form of column headers and/or short metadata descriptions of the dataset. The reason for this focus is that we are interested in enabling the training of scientific foundation models on diverse and multi-disciplinary scientific datasets, which currently is hindered by the fact that scientific data takes on a vast array of modalities (and, as new experiments are constructed, new modalities can be invented, too). It's therefore not straightforward to imagine how to train a foundation model on e.g. a mixture of not only images, videos, multi-dimensional timeseries, tabular data, point clouds, etc., but also be able to continually incorporate new modalities. An arbitrary modality, however, can be rendered as language, making language an appealing default modality to choose if a better encoding is not already known.
> - Cases where the specific number is important, e.g. small integers
> 	- Prompted by this comment, we tested xVal on a discrete dataset of small integers, even though this wasn't necessarily our intended use case. We generate 1 million examples of small integers sampled randomly and construct a dataset of 5 columns consisting of various functions of the small integers in the first column (e.g. x^3, 2x, etc.). We find that the resulting predictions are also quantized and achieve a median accuracy of 0.5% when reconstructing the values of the small integers. We believe this suggests that even if xVal were applied to cases where the specific number is important, it would do fairly well, though if flawless accuracy is required, it indeed might not be the best choice.
> - Tokenizing numerical strings of different formats (e.g. "1", "1.00", and "1E0")
> 	- When tokenizing numerical strings, we exclude any numbers occurring within quotation marks, which in our convention correspond to column headers or other forms of metadata. This means that even if we have a column called "planet_1", the 1 would not be tokenized with xVal. We use a regular expression (`pattern = r"(?<!\')-?\d+(\.\d+)?([eE][-+]?\d+)?(?!\'|\d)"`) designed to capture numbers in a variety of formats, including integers, floats, and scientific notation using a lowercase e or capital E.
> - On scientific notation and splitting the mantissa & exponent
> 	- You are correct that rendering numbers in scientific notation has been explored in a variety of ways for numerical tokenizations, both by keeping the number intact and by splitting the exponent and the mantissa and encoding them separately (e.g. 2109.03137), but in this case the mantissa is instead mapped onto a weighted average of a finite set of “prototype numerals”. Indeed, splitting the number in this way for xVal would alter the guarantee of numerical continuity, which is a core inductive bias that we want to preserve for our intended use cases.
> - Scaling & loss of precision
> 	- We address this by presenting the option of including multiple xVal tokens for varying scales. However, we don't yet have a solution that allows for arbitrary precision across all scales -- ultimately, a certain number of significant digits must be chosen to proceed with the xVal tokenization.

---

> > ### Author Response · Authors · 2025-02-22
> >
> > (continued)
> >
> > - Public Comment I think that the papers [2-5] presented in Alex Shtoff's comments are papers that should be cited in terms of encoding and handling numerical values. However, my review evaluates the paper from the perspective of numerical embedding for LLM, and I do not think that the novelty of the paper is lost by the fact that the mapping is an existing technology. Rather, I judged the weakness to be that the evaluation is inadequate as an LLM-required task.
> > 	- Thank you for this feedback. We will incorporate the papers cited in Alex Shtoff's comments into our introduction to have a more thorough discussion of previous work.
> > - Section 2.1: "Continuous embeddings have been previously proposed for use in attention mechanism in the context of speech recognition (Chorowski et al., 2014)." Since the relationship between the cited paper and continuous embeddings was unclear, it would be better to discuss in more detail which methods have already been proposed and whether they are similar or different ideas.
> > 	- Thank you for pointing out this sentence, which reads as a bit out-of-place in its current section. We will move it to the introduction instead, where we discuss other prior work, and will describe in more detail how it relates to the broader influences of xVal.
> >
> > **Optional comments:**
> > - How to tokenize numbers differently depending on their context (e.g. "4-dimensional spacetime", scientific constants, or different units)
> > 	- We agree that when handling large language corpuses, there are potentially multiple ways to tokenize numbers, or we might prefer to include multiple tokens corresponding to a given number (e.g. "2025", "3.14", etc.) depending on the context in which it appears. We are optimistic that continuous numerical encodings could co-exist with dedicated language encodings for particular numbers of interest. In our current setting, while our multi-scale extension of xVal allows for predictions across a wide range of numbers, it would currently be agnostic to the unit attached to that particular number at that particular scale (e.g. 1,000 meters and 1 km would be tokenized differently). It would be interesting to consider an extension of xVal in future work that includes minimal language tokens representing scientific units so that such encodings could be better addressed.
> > - This is not essential, but I am interested in the output of the number head corresponding to words that are not predicted to be of the [NUM] class (e.g., how do "desk", "chair", and "pen" correspond as numbers?) If there are any interesting trends, I would like to know about them.
> > 	- This is also an intriguing question! Unfortunately, given that our scope requires a limited language vocabulary, we are not able to properly investigate this question without greatly increasing our dataset sizes and compute resources. We thank the reviewer for this exciting proposition, however, and would be eager to consider this in future work.
> >
> > Thank you again for your thoughtful consideration of our work.
> >
> > Sincerely,
> > The Authors

---

### Review · Reviewer_uYAk · 2025-02-03

**Summary Of Contributions:**

The paper introduces xVal, a new continuous numerical tokenization method for scientific language models. Instead of representing numbers as sequences of text tokens, xVal embeds each number along a learnable direction in the embedding space, reducing token count to one per number and minimizing the vocabulary. Experiments on tasks like temperature forecasting and planetary orbit simulation demonstrate that xVal achieves superior out-of-distribution performance and computational efficiency compared to text-based encoding methods.

**Audience:**

Yes

**Claims And Evidence:**

Yes

**Requested Changes:**

- Incorporate relevant literature and discussions raised in the public comment by Alex Shtoff to better situate the work within the broader research context.
- Clarify the rationale for using language models for numerical-intensive tasks versus dedicated numerical models.
- If time permits, evaluate xVal in benchmarks that combine text and numerical data to assess its performance in more realistic scenarios.

**Strengths And Weaknesses:**

## Strengths:
- Authors present a novel, token-efficient method that improves numerical representation in language models.
- Authors provide comprehensive experiments across varied scientific datasets and arithmetic tasks.
- The experiments show strong out-of-distribution generalization and competitive in-distribution performance.
## Weaknesses:
- As stated in the public comment by Alex Shtoff, I agree that there is a line of research not included in the related works sections in this paper.
- Most experiments focus exclusively on numerical-intensive tasks, raising questions about the applicability of language models to these problems compared to dedicated numerical methods.
- It remains unclear to me how xVal would perform in more realistic settings where raw text and numbers coexist, such as in full scientific papers, compared to conventional tokenizer used by LLMs.

---

> ### Author Response · Authors · 2025-02-22
>
> Dear Reviewer uYAk,
>
> Thank you for your review! Please see our comments below.
>
> - Incorporate relevant literature and discussions raised in the public comment by Alex Shtoff to better situate the work within the broader research context.
> 	- Indeed, we will update the introduction of the paper to include a new paragraph in our introduction to cite the references that Alex Shtoff mentions.
> - Clarify the rationale for using language models for numerical-intensive tasks versus dedicated numerical models.
> 	- Numerical models can be powerful tools for understanding complex datasets, particularly if the underlying dynamics are somewhat well-understood. If we only wanted to predict e.g. orbital motion data, it will no question be optimal to use numerical solvers based on orbital mechanics instead of applying a Large Language Model. However, as we imagine the onset of scientific foundation models trained on a wide variety of data modalities and raw datasets from diverse sub-fields of science, we will need a more general solution. xVal is built from a premise in which language can be thought of as the "ultimate" modality for scientific data, as language is flexible enough to represent any dataset format. This language-first approach might be especially necessary to enable scientific foundation models, because in science, new data modalities are regularly created with the construction of new types of detectors and experiments. We therefore adapt the core structure of language embeddings to better represent numbers in hopes that this will be an important step in allowing for the training of truly interdisciplinary scientific foundation models. We will add some additional text in our introduction to further underscore this intended use case.
> - If time permits, evaluate xVal in benchmarks that combine text and numerical data to assess its performance in more realistic scenarios.
> 	- While evaluating the performance of xVal trained on a corpus of scientific papers is a very interesting direction, for the purposes of this work we intend xVal to be used to process numerically dense scientific datasets with minimal language included -- mostly in the form of short metadata descriptions of the data (e.g. detector conditions corresponding to the data represented) or column headers. This is because the problem we are primarily interested in solving is more about enabling training multi-disciplinary scientific foundation models on diverse datasets with arbitrary modalities (see previous response), and for this reason language is used because of its ability to represent arbitrary modalities. The fact that language as a modality also has the benefit of being able to incorporate significant natural language to represent e.g. scientific domain knowledge is an exciting direction that we leave for future expansions of this work.
>
> Sincerely,
> The Authors

---

> > ### Comment · Reviewer_uYAk · 2025-03-06
> >
> > Thank you for addressing my questions!
> >
> > > "Indeed, we will update the introduction of the paper to include a new paragraph citing the references that Alex Shtoff mentioned."
> >
> > Thank you for updating according to the common reply!
> >
> >
> > > "However, as we imagine the onset of scientific foundation models trained on a variety of data modalities and raw datasets from diverse sub-fields of science, we will need a more general solution."
> >
> > After the rebuttal, I still find two points unclear to me:
> > 1. Which approach is better—using an agent (e.g., an LLM-based system like AutoGPT) that calls a numerical solver when needed, or training a language model primarily on numerical data?
> > 2. (minor) Will changes in tokenization affect the outcomes of language model pre-training and fine-tuning? For example, could xVal negatively impact LLM pre-training?
> >
> > If the authors want to argue that their proposed method is a better pathway for training LLMs for solving numerical-intensive scientific tasks, I believe they should address these questions.
> >
> >
> > > "While evaluating the performance of xVal trained on a corpus of scientific papers is a very interesting direction, for the purposes of this work we intend xVal to be used to process numerically dense scientific datasets with minimal language included."
> >
> > I agree that benchmarking the proposed method on a scientific paper understanding task is unnecessary for this paper and its scope. It remains an interesting future direction and is not included in my current evaluation of the work.

---

### Review · Reviewer_Wd1e · 2025-02-07

**Summary Of Contributions:**

This paper introduces a continuous numerical encoding method for language models, embedding numbers directly in the embedding space via a learnable direction and a [NUM] token for single-token efficiency. A separate "number head," trained with MSE loss, ensures accurate numerical inference. The proposed methods reduces processing time, enhances interpolation and out-of-distribution generalization, and outperforms text-based encodings in scientific tasks like temperature forecasting and planetary orbit simulations. Additionally, multi-scale embeddings improve representation across diverse magnitudes. Authors address limitations of text-based encodings, offering a more efficient and accurate numerical reasoning framework for language models.

**Audience:**

Yes

**Claims And Evidence:**

Yes

**Requested Changes:**

To improve the effectiveness of the proposed approach, several refinements are recommended. First, replacing MSE loss with a combination of L1 and L2 loss or using negative log-likelihood (NLL) loss could enhance numerical accuracy. Additionally, adopting little-endian notation for numerical data could better align with the model’s left-to-right processing, and an empirical comparison between big-endian and little-endian representations would provide further validation. To enhance practical applicability, integrating symbolic computation or retrieval-based methods, such as Program-of-Thought prompting, and exploring program synthesis techniques could improve numerical robustness. Furthermore, evaluating the approach on real-world numerical datasets, such as those in finance, science, and engineering, while benchmarking against program-based prompting techniques, would strengthen its practical relevance. Finally, analyzing the model’s scalability to larger datasets and unseen numerical distributions, alongside a thorough error analysis, could help identify failure cases and guide future improvements.

**Strengths And Weaknesses:**

Strengths:

Novel Approach to handle numerical data, with single token for reduced computational cost and minimal vocabulary footprint.  Additionally, this inherently introduces end-to-end continuous mapping between input and output very necessary for time series data. This work also addresses  Text-based encodings  failure to interpolate property with their out-of-distribution performance.

Weaknesses:

As the authors as low mention that due to use of MSE-Loss, the numerical reconstruction is not guaranteed for sure. As MSE may predict averages, This limits the use case in real world scenarios.

To ensure complete accuracy when using a small transformer that decodes in a left-to-right manner, is'nt advisable to present numerical data in little-endian notation? This approach aligns with the natural processing direction of the model. Where as, humans do arithmetic in right to left, as this order aligns with the logical structure of conventional algorithms.

While the experiments demonstrated are relevant, they seem more theoretical than applicable to real-world complex tasks, especially when small language models with "Program-of-Thought" prompting perform exceptional with number and real-word data.

Overall the work is novel with fresh perspective, but the practicality in real-world as claimed by authors is very limited  given the efficacy of program based prompting techniques.

---

> ### Author Response · Authors · 2025-02-22
>
> Dear Reviewer Wd1e,
>
> Thank you for your thoughtful and careful review and for your suggestions for further improvement of our work! Below, we summarize our responses to each comment individually:
>
> - MSE loss vs. a combination of L1 and L2 loss
> 	- Thanks to your comment, we tested using Huber loss (a combination of L1 & L2 loss) instead of MSE loss within our numerical inference framework and saw that this tended to improve the numerical predictions in a number of different test cases. We will update the paper and code to indicate that this modification would be likely advisable for further improvements of the core results presented here.
> - "Additionally, adopting little-endian notation for numerical data could better align with the model’s left-to-right processing, and an empirical comparison between big-endian and little-endian representations would provide further validation."
> 	- This is a really interesting direction to consider. We did not benchmark against numerical tokenization strategies that use little-endian notation simply because they have not yet been commonly adopted for LLM training. There is recent work (e.g. 2403.05845) suggesting that little-endian notation could bring on the order of 10% improvements compared with the default tokenizations, so this benchmark could be interesting to include in future work if there is broader adoption in the LLM community. In the case of xVal, at least, changing the digit ordering shouldn't affect our performance as we process the entire number all at once.
> - Program-of-thought prompting
> 	- It's certainly true that recent LLMs have seen huge improvements in their numerical reasoning abilities in e.g. arithmetic tasks or mathematical word problems when Chain-of-Thought or Program-of-Thought prompting is introduced. However, we intentionally exclude such benchmarks because our intended use case is in predicting the behavior of variables within numerically dense scientific datasets for which the underlying dynamics might be unknown or poorly understood. In these scenarios, natural language reasoning (as in Chain-of-Thought) will be unhelpful due to the small amount of natural language data present, and even Program-of-Thought reasoning will be limited in its applicability since the correct formula(s) for predicting the behavior of the system might not be known.
> - "Furthermore, evaluating the approach on real-world numerical datasets, such as those in finance, science, and engineering, while benchmarking against program-based prompting techniques, would strengthen its practical relevance."
> 	- While one of our main datasets indeed consists of somewhat simplified 2D planetary orbit simulations, our other main dataset (ERA5) is a large-scale reanalysis dataset that incorporates hourly real-world meteorological observations since 1979.
> - "Finally, analyzing the model’s scalability to larger datasets and unseen numerical distributions, alongside a thorough error analysis, could help identify failure cases and guide future improvements."
> 	- As we detail in Appendix D.1.1 and D.2, we use 1.25 million examples each for the ERA5 and planetary motion datasets, which we believe is sufficiently large-scale to illustrate the future utility of xVal in a variety of nontrivial contexts. Moreover, we illustrate out-of-distribution numerical generalization for xVal in Figure 8 in two examples and include errors for each result in our tables. Please let us know if you have suggestions for additional error analyses that would help support our results.
>
> Thanks again,
> The Authors

---

### Public Comment · ~Alex_Shtoff1 · 2025-01-22
**Overlooked literature**

The basic idea in this paper is, essentially, mapping a numerical feature $x$ to a point on a parametric curve in the embedding space, using a linear combination of vectors, where the coefficients come from some basis (Equation 1). In general, such a curve has the form:
$$
C(x) = \sum_{i=1}^K \mathbf{v}_i B_i(x).
$$
In Equation 1 we have $K=2k+1$ basis functions which are just scaled hyperbolic tangents. When $k=0$, we have a single vector, and it reduces to mapping $x$ to a point on a _line_ in the embedding space. In this sense, when $k=0$, the hyperbolic tangent is just a feature normalization technique.

It appears this paper re-invents a wheel, does not cite the huge amount of previous work, and does not explain what differentiates the wheel proposed in this paper from the existing wheels. The idea itself of parametrizing curves using a well-designed basis for use on a computer is not new, and dates back to Bézier curves [1] that are extensively used in computer graphics and computer aided design, and probably many other fields. In fact, all the fonts we read on our screens while reviewing the paper are drawn using Bezier curve.  Bézier also coined the term  "control points" for the vectors $\mathbf{v}_1, \dots, \mathbf{v}_k$ above.  The fact that in computer graphics the vectors are in $\mathbb{R}^3$ or $\mathbb{R}^2$ instead of some high-dimensional embedding space $\mathbb{R}^d$, and that the basis are Bernstein polynomials rather than scaled $\tanh$ functions, does not change the essence of the idea. There is an enormous amount literature that builds upon Bézier's idea in the computer graphics community.

The idea has also been employed in machine learning as well, especially in the recommender system community. Here, similarly to what this work proposes, the control points are learned. The IJCAI'17 paper by Guo et. al. [3] directly proposes the idea of mapping a numerical value to a point on a line, similar to this work. The NeurIPS'22 work by Gorishny et. al [2] uses a first-order B-Spline basis to parametrize a piecewise-linear curve. In the AAAI'24 paper by Rügamer et. al. [4], and a TMLR'24 paper by Shtoff et. al. [5], the proposed basis is a higher order B-Spline basis. In these two works the model consuming the embedding vectors come from the family of factorization machines, rather than neural networks, but the idea of parametrizing a curve in the embedding space using a basis stands.

I believe this paper should, as a minimum, state the fact that the idea is not new by citing at least one survey work from computer graphics, give credit to Pierre Bézier, cite the papers from the ML community, and explain how this work extends these existing ideas: scaled $\tanh$ functions to support multiple feature scales, the consumer of the embeddings is an LLM rather than recommendation system models, maybe a different potential application, etc... A discussion of *why* the scaled $\tanh$ basis is useful in the context of this work, rather than the others that already exist in the literature, is also important.

It could, of course, be improved by trying out various bases in practice, or discussing their theoretical properties. But this is, of course, a decision of the authors, since even without it I believe the work is already interesting.

Since one of the overlooked papers from machine learning I pointed above is mine, I did not volunteer to review the paper. I have a personal interest here, and I believe it would be unethical of me to make acceptance recommendations. But I do hope the reviewers find this comment valuable.


---
[1]: Bézier, Pierre E. How Renault uses numerical control for car body design and tooling. No. 680010. SAE Technical Paper, 1968.

[2]: Gorishniy, Y., Rubachev, I., & Babenko, A. (2022). On embeddings for numerical features in tabular deep learning. Advances in Neural Information Processing Systems, 35, 24991-25004.

[3]: Guo, H., Tang, R., Ye, Y., Li, Z., & He, X. (2017, August). DeepFM: a factorization-machine based neural network for CTR prediction. In Proceedings of the 26th International Joint Conference on Artificial Intelligence (pp. 1725-1731).

[4]: Rügamer, D. (2024, April). Scalable Higher-Order Tensor Product Spline Models. In International Conference on Artificial Intelligence and Statistics (pp. 1-9). PMLR.

[5]: Shtoff, A., Abboud, E., Stram, R., & Somekh, O. (2024, December) Function Basis Encoding of Numerical Features in Factorization Machines. Transactions on Machine Learning Research.

---

> ### Author Response · Authors · 2025-02-22
>
> Dear Alex,
>
> Thank you for your interest in our paper and for sharing this additional context. We agree with our reviewers that citing the works you include in your comment would be an asset to our paper, and we will be sure to include these citations in a new paragraph in our introduction to provide a richer backdrop for the influences that led to the development of xVal as a numerical tokenization scheme within a language model framework.
>
> Sincerely,
> The Authors

---

### Comment · Action_Editor_5SSY · 2025-02-07
**Start discussions**

Dear authors,

You have three reviews and one public comment. Please address their comments as soon as possible.
The discussion period will end on February 21st, 2025.

---

Dear reviewers,

Thank you for submitting your reviews.
As this paper received three reviews, now it's time to have discussions with authors.

You will be able to submit your formal decision recommendation starting in 2 weeks.
Your prompt responses to authors' comments would be very appreciated.

Remember that different from other journals / conferences, [TMLR's acceptance criteria](https://jmlr.org/tmlr/acceptance-criteria.html) are based on positive answers to the following two questions
1. Are the claims made in the submission supported by accurate, convincing and clear evidence?
2. Would some individuals in TMLR's audience be interested in the findings of this paper?

It's still not too late to review the [TMLR's reviewer guideline](https://jmlr.org/tmlr/reviewer-guide.html) in case you are not aware of that.

Thank you!
Your AE

---

### Decision · Action_Editor_5SSY · 2025-03-16

**Recommendation:** Reject

**Comment:**

The final recommendations are two "Leaning Reject"s and one "Leaning Accept". All reviewers agree that this work needs more revisions while some of them found some novelty in this work.

While the novelty of the method is not that important at TMLR, the reviewers seem leaning to agree some novelty of the method and I find the claim is partially supported by the evaluation in the current revision.

As suggested by Reviewers 116U and uYAk, this paper needs more evaluations and discussions to support its claims, including experiments using conventional tokenization and prior work that the public comment and some of the reviewers pointed out that this paper missed.

Based on TMLR's acceptance criteria, I want to suggest that the paper makes claims more specific and rather than make the claims look broad and significant as TMLR values technical soundness as well as clarity of the narrative and arguments. For instance, if the paper shows experimental results that the proposed method works better than existing approaches for specific tasks, the paper should make claims specific like "We demonstrate that our method outperformed existing methods (X, Y, Z ) for tasks A, B, and C."

With proper claims and more convincing evidence to support the claim, this paper may be accepted to TMLR. But it still requires significant revisions and I strongly suggest that the authors to refer to the reviewers' suggestions and public comment to improve their work before they resubmit this work to TMLR (or somewhere else).

**Audience:**

Continuous numerical tokenization is an interesting, challenging NLP problem and some of TMLR's audience would be interested in the discussions

**Claims And Evidence:**

This work introduce xVal, a novel encoding method for numerical values in language models specialized for scientific data analysis, claiming that the method is token-efficient and requires a minimal vocabulary footprint.
As the public comment and Reviewers 116U & uYAk suggested, [1]-[5] should be properly cited and compared against the proposed method. Reviewer 116U also points out that the lack of the evaluation as a weakness.

Reviewers uYAk also points out that its current evaluations are not convincing enough to claim that the proposed method is better for training LLMs for solving numerical-intensive scientific tasks (See [the follow-up comment](https://openreview.net/forum?id=kSPHESKMve&noteId=ZuK8mvoHl5)). Although the reviewer considers the second question in the follow-up comment, I think that it is indeed critical to discuss the second point as this paper claims that "xVal consistently provides better interpolation properties and is more compute-efficient than prior work" without mentioning specific conditions where the method can work better than prior ones. If this is not the intention, the claim should be more specific and modified accordingly.


[1]: Bézier, Pierre E. How Renault uses numerical control for car body design and tooling. No. 680010. SAE Technical Paper, 1968.

[2]: Gorishniy, Y., Rubachev, I., & Babenko, A. (2022). On embeddings for numerical features in tabular deep learning. Advances in Neural Information Processing Systems, 35, 24991-25004.

[3]: Guo, H., Tang, R., Ye, Y., Li, Z., & He, X. (2017, August). DeepFM: a factorization-machine based neural network for CTR prediction. In Proceedings of the 26th International Joint Conference on Artificial Intelligence (pp. 1725-1731).

[4]: Rügamer, D. (2024, April). Scalable Higher-Order Tensor Product Spline Models. In International Conference on Artificial Intelligence and Statistics (pp. 1-9). PMLR.

[5]: Shtoff, A., Abboud, E., Stram, R., & Somekh, O. (2024, December) Function Basis Encoding of Numerical Features in Factorization Machines. Transactions on Machine Learning Research.

**Resubmission Of Major Revision:**

The authors may consider submitting a major revision at a later time.